Methods

# Reversible and effective cell cycle synchronization method for studying stage-specific processes

Yu-Lin Chen[1], Syon Reddy[1], Aussie Suzuki[1,2]

**The cell cycle is a crucial process for cell proliferation, differentiation, and development. Numerous genes and proteins play pivotal roles at specific cell cycle stages to ensure precise regulation of these events. Understanding the stage-specific regulations of the cell cycle requires the accumulation of cell populations at desired cell cycle stages, typically achieved through cell cycle synchronization using kinase and protein inhibitors. However, suboptimal concentrations of these inhibitors can result in inefficiencies, irreversibility, and unintended cellular defects. In this study, we have optimized effective and reversible cell cycle synchronization protocols for human RPE1 cells by combining high-precision cell cycle identification techniques with high-temporal resolution live-cell imaging. These reproducible synchronization methods offer powerful tools for dissecting cell cycle stage-specific regulatory mechanisms.**

## Introduction

The cell cycle is precisely regulated by a variety of kinases and proteins, with checkpoint mechanisms overseeing each stage to ensure proper cell cycle progression (Schafer, 1998; Vermeulen et al, 2003; Harper & Brooks, 2005). Disruption of this regulatory system can result in cancer and developmental diseases (Matthews et al, 2022). The cell division cycle includes four major stages: G1, S, G2, and M phases, each with distinct functions. During the G1 phase, cells express proteins necessary for DNA synthesis, preparing for entry into the S phase. Cyclin D, in conjunction with Cdk4/6, plays a critical role in this process. The Cyclin D-Cdk4/6 complex phosphorylates the retinoblastoma protein (Rb), facilitating the release of Rb from E2F, an essential transcription factor (Schafer, 1998; Vermeulen et al, 2003; Harper & Brooks, 2005; Narasimha et al, 2014). This promotes E2F-dependent gene expression, including that of Cyclin E and Cyclin A, leading to the S phase entry. During the S phase, DNA polymerases orchestrate DNA replication. Cyclin E-Cdk2 promotes the transcription of histones, which are required

for forming nucleosomes upon DNA synthesis (Schafer, 1998; Vermeulen et al, 2003; Harper & Brooks, 2005; Armstrong et al, 2023). After completing DNA replication, cells enter the G2 phase. The G2/M transition requires the activation of Cyclin B-Cdk1, and proper mitotic progression necessitates the degradation of Cyclin B (Schafer, 1998; Vermeulen et al, 2003; Harper & Brooks, 2005). The M phase, known as mitosis, includes five substages: prophase, prometaphase, metaphase, anaphase, and telophase (Iemura et al, 2021).

Accumulating a cell population at the desired cell cycle stage is crucial for studying and identifying stage-specific gene/protein functions and interactions. One primary method for achieving this is fluorescence-activated cell sorting (FACS). FACS can sort cells based on specific cell cycle markers or DNA content in both live and fixed cells (Juan et al, 2002; Van Rechem et al, 2021). However, this technique requires specialized FACS equipment and a large number of cells, particularly when targeting low-abundance cell cycle stages, such as mitotic cells, in asynchronous populations (Whetstine & Van Rechem, 2022). Moreover, FACS often struggles to distinguish between the G2 and M phases and to identify detailed substages within other cell cycle stages. Another widely used method involves cell cycle kinase and protein inhibitors (Banfalvi, 2011; Hadfield et al, 2022; Wang, 2022). For example, Cdk4/6 inhibitors are extensively used in both basic research and clinical therapy for breast cancer, effectively arresting cells in the G1 phase (Wang et al, 2024). DNA polymerase inhibitors and DNA damage agents can arrest cells in the S phase, whereas Cdk1 inhibitors can halt cells in the G2 phase. Microtubule inhibitors are commonly used to synchronize cells in mitosis (Ligasova & Koberna, 2021). Although these cell cycle inhibitors are effective and user-friendly, it is crucial to use optimal concentrations and treatment durations. Using concentrations lower than optimal can lead to slower cell cycle progression with unintended defects, whereas higher concentrations can cause irreversible effects on the cell cycle. Both scenarios can potentially produce artificial results in experiments.

In this study, we carefully evaluate the effectiveness of widely used inhibitors for cell cycle synchronization at each stage of the cell cycle (G0, G1, S, G2, and M phases). These synchronization protocols were specifically optimized for the hTERT-immortalized

[1]McArdle Laboratory for Cancer Research, Department of Oncology, University of Wisconsin-Madison, Madison, WI, USA [2]Carbone Comprehensive Cancer Center, University of Wisconsin-Madison, Madison, WI, USA

Correspondence: aussie.suzuki@wisc.edu

retinal pigment epithelial cell line (RPE1), a widely used, non-transformed human epithelial cell line in diverse research fields. By integrating a recently developed immunofluorescence (IF)-based cell cycle identification method (Chen et al, 2024b) with high-temporal resolution live-cell imaging, we provide a comprehensive analysis of the impact of cell cycle arrest induced by major cell cycle inhibitors and their reversibility. The optimized cell synchronization techniques and thorough evaluation presented in this study will be invaluable for investigating stage-specific regulatory mechanisms of the cell cycle.

# Results

### Cell cycle synchronization in G1 phase

We initially determined the detailed distribution of cell cycle phases in asynchronous RPE1 cells, which served as the standard in this study, using a recently developed high-precision, ImmunoCellCycle-ID method (Chen et al, 2024b) (Fig S1A and B). Briefly, cells were fixed and stained during the logarithmic growth phase (see the Materials and Methods section). An advantage of the use of this IF-based cell cycle identification method allows us to determine detailed substages in cell cycle: G1, early/middle S, late S, early/middle G2, late G2, and each stage of mitosis, with a single cell resolution and accuracy. ImmunoCellCycle-ID uses antibodies against PCNA, CENP-F, and CENP-C to precisely identify cell cycle phases. PCNA is widely used as a marker of S phase, as it exhibits a distinct punctate nuclear pattern during this phase (Schonenberger et al, 2015; Chen et al, 2024b). However, it cannot differentiate between G1 and G2 phases because PCNA displays a uniform nuclear distribution in both. CENP-F, a well-characterized kinetochore-associated protein, localizes to kinetochores from prophase to anaphase during mitosis (Ciossani et al, 2018; Wynne & Vallee, 2018). Notably, CENP-F begins to accumulate in the nucleus specifically from the S phase and remains nuclear-localized until late G2 phase (Liao et al, 1995; Hussein & Taylor, 2002; Loftus et al, 2017). By combining CENP-F and PCNA staining, we can accurately distinguish G1, early/mid-S, late S, and G2 phases. CENP-C, a constitutive centromere-associated network protein, remains kinetochore-localized throughout the cell cycle (Musacchio & Desai, 2017; Chen et al, 2024a Preprint). It serves as a robust marker for validating cell cycle stages and specifically distinguishing early/mid-G2 from late G2 phases. CENP-C fluorescence intensifies during S phase, peaks in early G2, and resolves into discrete pairs in late G2, reflecting the formation of new kinetochores on newly synthesized centromeric DNA (Gascoigne & Cheeseman, 2013; Chen et al, 2024a Preprint, 2024b). Our results revealed that ~50% of the cells were in the G1 phase, 20% in the early/middle S phase, 10% in the late S phase, 11% in the early/middle G2 phase, 4% in the late G2 phase, and 5% in mitosis (Fig S1B), aligning with previous results (Lau et al, 2009; McKinley & Cheeseman, 2017; Pei et al, 2022; Chen et al, 2024b).

Effective and reversible cell cycle synchronization is crucial for studying protein functions associated with the cell cycle. This synchronization is typically achieved using chemical inhibitors that

target kinase activities or essential proteins required for cell cycle progression (Mills et al, 2017; Wang, 2022). The double thymidine block is a traditional method for synchronizing cells in the G1 phase (Chen & Deng, 2018). Briefly, cells are incubated with 2 mM thymidine for approximately 16 h. After a 9-h washout period, a second round of thymidine treatment is applied for ~16 h. Our analysis confirmed that after the second round of thymidine treatment, ~70% of cells were successfully arrested in the G1 phase (Fig S1C and D). However, we observed that about 30% of cells were in the S and G2 phases (Fig S1C and D). As expected, the short-time treatments of thymidine, 6- and 12-h intervals during the first round, did not result in significant G1 phase accumulation (Fig S1D). Whereas the double thymidine block is a useful and effective method for the synchronization in G1 phase, it is time-intensive, highlighting the need for a simpler and more user-friendly alternative.

Cyclin D, in conjunction with Cdk4/6, plays a pivotal role in regulating the G1 phase of cell cycle progression (Fassl et al, 2022). Previous research has demonstrated that Cdk4/6 inhibitors can induce G1 phase arrest in a wide variety of cells (Knudsen et al, 2020; Trotter & Hagan, 2020; Jost et al, 2021; Pennycook & Barr, 2021). Consequently, we investigated the detailed effects of palbociclib, a highly selective Cdk4/6 inhibitor, on G1 phase synchronization (Liu et al, 2018). Prior studies have indicated that cells exposed to elevated concentrations of palbociclib fail to resume cell cycle progression after washout (Trotter & Hagan, 2020). Therefore, we tested five concentrations of palbociclib: 0.05, 0.1, 0.25, 0.5, or 1 $\mu$M. After treating cells with these concentrations of palbociclib for 24 h, they were subsequently subjected to the immunofluorescence-based cell cycle measurements (Fig 1A). Our findings revealed that almost 100% of the cells treated with palbociclib were arrested in G1 phase across a range of concentrations from 0.1 to 1 $\mu$M (Fig 1B). However, when treated with 0.05 $\mu$M of palbociclib, over 25% of the cells entered S phase, suggesting that this concentration is insufficient to fully arrest cells in G1 phase. We next investigated whether cells treated with palbociclib could resume cell cycle progression after a washout. For this purpose, cells treated with palbociclib for 24 h were subjected to a washout process using culture media supplemented with STLC, an Eg5 inhibitor that induces mitotic arrest to prevent progression to the next cell cycle stage. 18 h post-washout, the cells were fixed and assessed the cell cycle distribution (Figs 1C and S2A). Our findings revealed that cells treated with concentrations ranging from 0.05 to 0.5 $\mu$M of palbociclib demonstrated a 50–60% incidence of the S phase and up to 20% of cells in mitosis, suggesting that these concentrations enable the resumption of cell cycle progression. However, ~30% of cells treated with these concentrations remained arrested in the G1 phase. In contrast, treatment with 1 $\mu$M palbociclib resulted in a significantly higher proportion of cells in the G1 phase (~55%), indicating an impaired ability to restart cell cycle progression at this concentration. Similar results were observed after the washout of a double thymidine block. 12 h after the second thymidine treatment washout, ~30% of the cells remained in the G1 phase (Fig S1D). To further explore why ~30% of cells were arrested in the G1 phase after palbociclib washout, we quantified the proportion of cells in the G0 phase. This analysis was performed on cells treated with palbociclib (0.25 and 1.0 $\mu$M) for 24 h, as well as 24 h post-washout, using the method detailed in our recent study (Chen et al, 2024b)

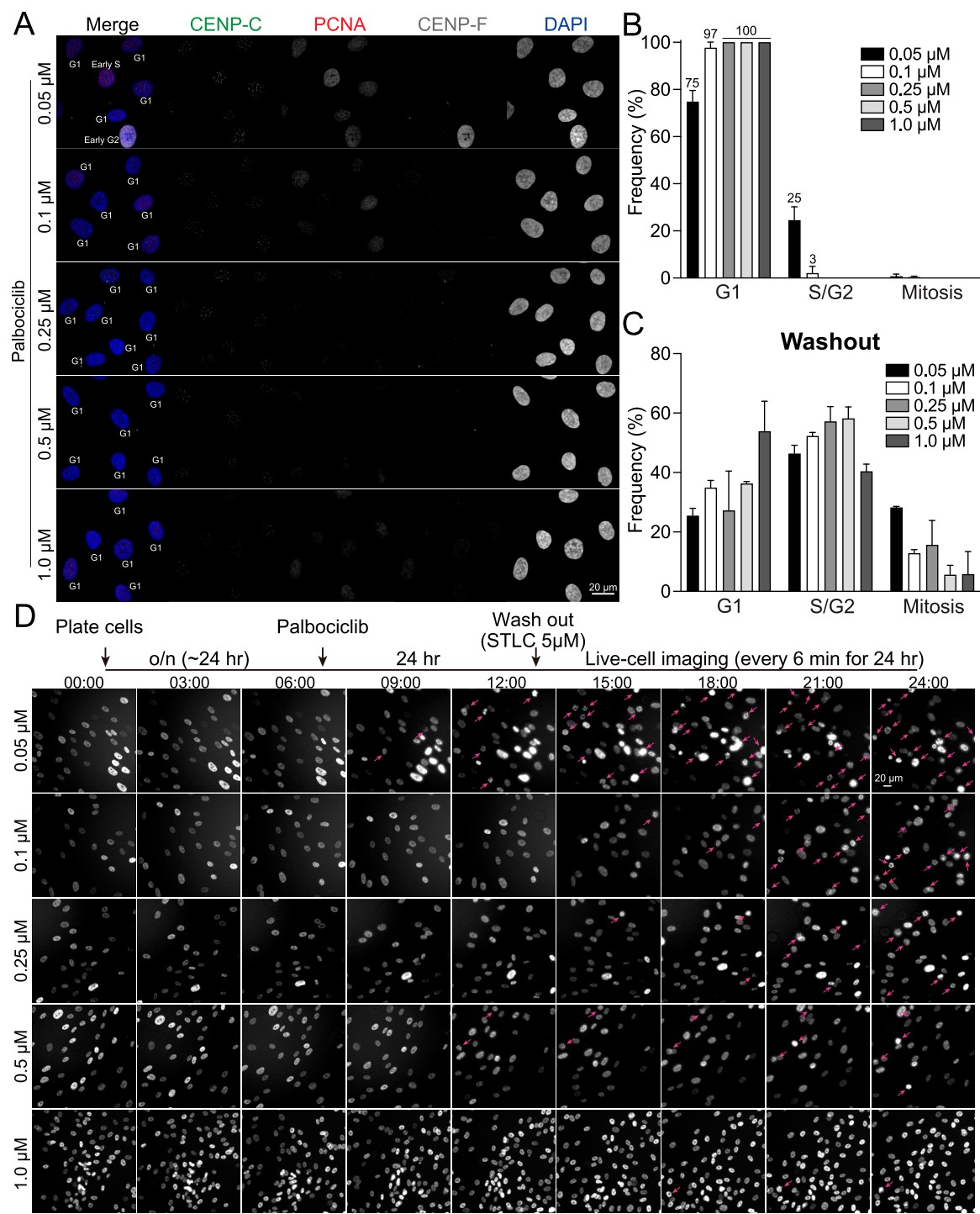

**Figure 1. G1 phase synchronization and release by palbociclib.**
**(A)** Representative immunofluorescence images of RPE1 cells treated with palbociclib (0.05, 0.1, 0.25, 0.5, or 1 $\mu$M) for 24 h, labeled with antibodies for CENP-C, PCNA, and CENP-F. **(A, B)** Proportion of G1, S/G2, or M phase in condition (A). From left to right, n = 416, 408, 383, 417, 487 (from two replicates). **(C)** Proportion of in G1, S/G2, or M phase, analyzed at 18 h after the washout of palbociclib by 5 $\mu$M of STLC containing media. From left to right, n = 369, 317, 336, 355, 393 (from two replicates). **(D)** Schematic timeline of live-cell imaging sequence (top). Representative live-cell imaging of H2B-GFP expressing RPE1 cells treated with palbociclib (0.05, 0.1, 0.25, 0.5, or 1 $\mu$M for 24 h) (Bottom). Palbociclib was washed out before imaging in 5 $\mu$M of STLC containing media. Mitotic cells are indicated with pink arrows. Imaging was performed on at least two independent replicates.

(Fig S2B and C). We observed that 9–13% of cells treated with palbociclib for 24 h and 4–7% of cells post-washout were in the G0 phase. These findings suggest that a small subset of G1-phase cells (~10%) enters the G0 phase during 24 h of palbociclib treatment, with a fraction of these G0-phase cells remaining in G0 phase even after the washout. To validate these results, we performed live-cell imaging of RPE1 H2B-EGFP cells immediately after palbociclib washout, using STLC-supplemented culture media (Fig 1D). In alignment with the immunofluorescence quantifications, cells exposed to palbociclib at concentrations ranging from 0.1 to 0.5 $\mu$M entered mitosis ~12–15 h post-washout (Fig 1D, arrows). Conversely, cells treated with 0.05 $\mu$M palbociclib exhibited mitotic cells as early as 9 h after washout, whereas those treated with 1 $\mu$M rarely showed signs of mitosis. We evaluated the efficacy of G1 phase synchronization using additional Cdk4/6 inhibitors, abemaciclib and ribociclib (Mills et al, 2017; Trotter & Hagan, 2020). Among the tested inhibitors, palbociclib demonstrated superior potency, achieving complete G1 phase arrest (100%) at concentrations >0.1 $\mu$M in RPE1 cells. In comparison, abemaciclib and ribociclib, at concentrations of 2 $\mu$M, arrested ~70% and 90% of cells in the G1 phase, respectively (Fig S2D and E). To summarize, our study suggests that palbociclib concentrations ranging 0.1–0.5 $\mu$M, which effectively induce G1 phase arrest, allow cells to resume cell cycle progression following washout in RPE1 cells.

## Cell cycle synchronization in G0 phase

The G0 phase, known as the resting phase, is a state in which cells do not undergo division until they re-enter the G1 phase. To synchronize cells in the G0 phase, we used the traditional serum starvation method, wherein cells were incubated in serum-free media (Kim et al, 2016). RPE1 cells were maintained in this condition for 48 h, as a 24-h incubation period was insufficient to induce a robust G0 phase arrest. Accurate detection of the G0 phase was achieved using the optimized ImmunoCellCycle-ID method (Chen et al, 2024b). Briefly, cells were co-labeled with PCNA and acetylated tubulin to identify primary cilia, a key marker of the G0 phase (Santos & Reiter, 2008; Chen et al, 2024b) (Fig S3A and B). PCNA was used to distinguish between G0 and S phases, as a subset of S phase cells also exhibit cilium formation (Spalluto et al, 2013; Ford et al, 2018). This serum starvation protocol resulted in ~80% of cells entering the G0 phase (Fig S3A–C). After re-incubation in serum-containing media, the proportion of G0 phase cells decreased to ~50% after 24 h and ~20% after 48 h. These findings confirm that serum starvation is an effective method for synchronizing cells in the G0 phase in RPE1 cells; however, G0-synchronized cells exhibit a slower re-entry into the cell cycle upon serum reintroduction.

## Cell cycle synchronization in S phase

To achieve S phase synchronization, we evaluated the efficacy of both aphidicolin and hydroxyurea (HU) treatments. Aphidicolin, a tetracyclic diterpene antibiotic, specifically inhibits DNA polymerases, enzymes essential for DNA replication during the S phase (Ikegami et al, 1978; Krokan et al, 1981). The effect of aphidicolin on cell cycle progression has been a subject of debate, with varying studies presenting contradictory findings. Some research posits

that aphidicolin induces an arrest in the early S phase (Bhaud et al, 2000; Xu et al, 2001, 2011; Maeda et al, 2014; Mazouzi et al, 2016; Fragkos et al, 2019), whereas others suggest it causes cells to halt at the G1 phase, likely right on the cusp of the G1-S transition (Saintigny et al, 2001; Engstrom & Kmiec, 2008; Szczepanski et al, 2019; Yiangou et al, 2019). Hydroxyurea (HU), a potent inhibitor of ribonucleotide reductase, is widely used to arrest cells in the S phase (Spalluto et al, 2013). By inducing replication stress during early S phase, it effectively inhibits DNA replication (Musialek & Rybaczek, 2021).

To elucidate the precise impact of aphidicolin on cell cycle progression, we conducted immunofluorescence-based cell cycle analysis using RPE1 cells. Our experiments involved a 24-h treatment with aphidicolin at concentrations of 2.5, 5, or 10 $\mu$g/ml. We found that ~90% of aphidicolin-treated cells showed an absence of punctuate PCNA signals in the nucleus, similar to the pattern observed in G1 or G2 phase cells (Figs 2A and B and S4A). Unexpectedly, some of these cells displayed nuclear CENP-F signals, characteristic of G2 phase cells, albeit at relatively weaker intensities, similar to S-phase cells (Fig 2A). This observation led us to hypothesize that aphidicolin suppresses PCNA punctate nuclear patterns, a hallmark of S phase, when arresting cells in the S phase. To test this hypothesis, we analyzed additional markers, BRCA1 and phospho-Rb (pRb). We examined the spatiotemporal dynamics of BRCA1 during cell cycle progression, co-staining with PCNA and CENP-C or CENP-F to accurately determine cell cycle stages in asynchronous RPE1 cells. In G1 phase, BRCA1 signals were absent in the nucleus (Fig S4A) (Feng et al, 2013). In early/mid-S phase, BRCA1 formed distinct nuclear foci, which significantly increased during late S phase. By early G2 phase, the number of BRCA1 nuclear foci markedly decreased. Consequently, most BRCA1-positive cells (>90%) were in either S phase (~59%, characterized by PCNA punctate signals and CENP-F nuclear signals) or G2 phase (~32%, characterized by significantly brighter CENP-F nuclear signals or brighter/paired CENP-C kinetochore signals) (Fig S4A–C). In contrast, most BRCA1-positive cells treated with 5 $\mu$g/ml of aphidicolin for 24 h lacked punctate PCNA nuclear signals but showed weaker CENP-F nuclear signals compared with G2 phase cells. These findings indicated that these cells were in the S phase, despite the absence of the typical PCNA punctate nuclear signals. Based on BRCA1 and CENP-F staining, ~72% of cells were arrested in S phase, whereas 28% were in G1 phase following 24-h treatment with 5 $\mu$g/ml aphidicolin (Fig S4A–C). To further validate these results, we assessed pRb, PCNA, and CENP-F staining. Our recent study demonstrated a significant increase in nuclear pRb signals from early S phase (Fig S4C–E) (Chen et al, 2024b). In asynchronous RPE1 cells, we found that ~90% of pRb-positive cells exhibited either exclusive nuclear CENP-F signals (indicating G2 phase) or both punctate nuclear PCNA and CENP-F signals (indicating S phase). pRb-positive cells lacking both PCNA nuclear puncta and nuclear CENP-F signals were likely in late G1 phase (Spencer et al, 2013; Arora et al, 2017); however, they comprised only ~10% of the total interphase population. Similar to BRCA1, ~80% of pRb-positive cells in asynchronous RPE1 cells displayed punctate nuclear PCNA and nuclear CENP-F signals, indicating them as S phase cells, whereas ~20% exhibited CENP-F nuclear signals without PCNA puncta,

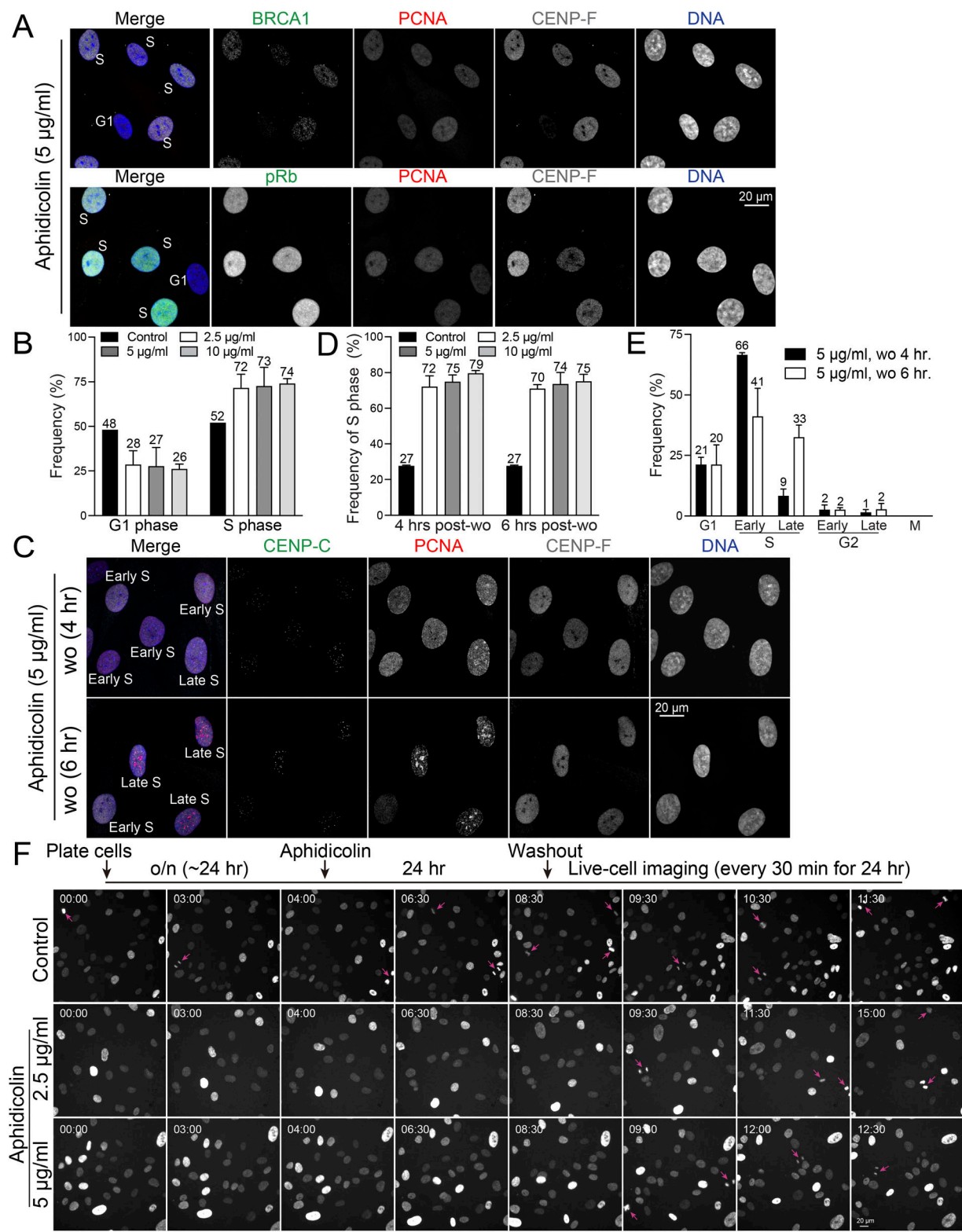

**Figure 2. S phase synchronization and release in RPE1 cells using aphidicolin.**
**(A)** Representative immunofluorescence images of RPE1 cells treated with 5 µg/ml aphidicolin for 24 h, labeled with antibodies for BRCA1 or pRb, PCNA, and CENP-F.
**(B)** Proportion of G1 or S phase in RPE1 cells treated with aphidicolin at concentrations of 2.5, 5, or 10 µg/ml for 24 h. From left to right, n = 424, 404, 379, 419 (from two replicates). Data represented from two experimental replicates. **(C)** Representative immunofluorescence images of RPE1 cells after 4 or 6 h post-aphidicolin washout (5 µg/ml for 24 h), labeled with antibodies for CENP-C, PCNA, and CENP-F. **(D)** Proportion of cells in S phase after 4 or 6 h post-aphidicolin washout. From left to right, n = 424, 379, 350, 341, 424, 401, 375, 413 (from two replicates). **(E)** Proportion of cells at different cell cycle stages (G1, Early S, Late S, Early G2, Late G2, and Mitosis) after 4 or 6 h post-

suggesting G2 phase. Notably, in cells treated with aphidicolin, most pRb-positive cells (~78%) lacked punctate PCNA nuclear signals but exhibited weaker CENP-F nuclear signals compared with typical G2 phase cells, indicating that these cells were in the S phase (Fig S4C–E). Unlike asynchronous cells, no pRB-positive cells lacking both PCNA and CENP-F signals were observed. In summary, based on pRb and CENP-F staining, ~87% of cells were arrested in the S phase after 24 h of treatment with 5 µg/ml aphidicolin. Furthermore, based on CENP-F and BRCA1 or pRb staining, we concluded that aphidicolin at concentrations of 2.5, 5, and 10 µg/ml effectively arrested cells in the S phase likely near the G1/S boundary, achieving an arrest rate of 70–80% (Fig 2A and B). Consistent with these findings, live-cell imaging revealed that cells treated with aphidicolin at concentrations of 2.5 or 5 µg/ml did not exhibit any mitotic entry after 9 h of treatment, whereas control cells continued to enter mitosis within the 24-h imaging period (Fig S5A). These results suggest that whereas PCNA staining or DNA content-based assays may indicate G1 phase arrest, the inclusion of additional markers such as CENP-F, BRCA1, and pRb reveals that aphidicolin predominantly arrests cells in the very early S phase.

To achieve better S phase synchronization, we aimed to determine the timing and conditions under which cells could progress through the S phase after the removal of aphidicolin. For this purpose, we incubated cells with aphidicolin at concentrations of 2.5, 5, or 10 µg/ml for 24 h, and subsequently fixed and stained the cells at 4 or 6 h after removing aphidicolin. Our results revealed that ~80% of the cells exhibited PCNA nuclear puncta, enabling the identification of early/middle and late S phase stages at both 4 and 6 h after aphidicolin removal across all tested concentrations (Figs 2C and D and S5B). Specifically, at 4 h post-aphidicolin washout at a concentration of 5 µg/ml, ~66% of cells were in early S phase and 9% were in late S phase (Fig 2E). This shifted to 41% in early S phase and 33% in late S phase by 6 h (Fig 2E). Similar trends were observed in cells treated with 2.5 or 10 µg/ml at 4 or 6 h after removal of aphidicolin (Fig S5B). These observations demonstrate a dynamic recovery, with about 80% of RPE1 cells successfully progressing to the S phase within 4–6 h after a 24-h exposure to aphidicolin at concentrations ranging from 2.5 to 10 µg/ml. To further validate these results, we conducted live-cell imaging after aphidicolin washout (Fig 2F). Mitotic cells appeared only 9 h after aphidicolin washout, whereas control cells continued to exhibit mitotic cells during live imaging (Fig 2F, arrows). This corresponds to the results obtained from the fixed immunofluorescence-based cell cycle analysis (Fig 2A–E).

Another widely used method for synchronizing cells in the S phase is HU treatment (Musialek & Rybaczek, 2021). Exposure to 2 mM HU for 24 h resulted in ~51% of cells being arrested in the S phase, with the majority (~45%) in early S phase and a smaller fraction (~6%) in late S phase (Fig S5C and D). Unlike aphidicolin treatment, HU-treated cells retained characteristic punctate PCNA nuclear signals of S phase. Although HU demonstrates lower efficiency in synchronizing cells in the S phase compared with aphidicolin washout, it offers the advantage of not requiring a washout step, making it potentially useful for traditional early S phase synchronization.

In conclusion, our study not only dissects the cell cycle arrest induced by aphidicolin but also highlights its capability for effective S phase synchronization upon washout. Aphidicolin removal is effective for studies focusing on early S phase within 4 h, and on late S phase after 6 h or more.

### Cell cycle synchronization in G2 phase

The Cyclin B-Cdk1 complex orchestrates both mitotic entry and exit. To initiate mitosis, Cyclin B-Cdk1 must be activated by Cdc25 phosphatase, which dephosphorylates Cdk1, converting it from its inactive to active form (Vassilev, 2006). Inhibition of Cdk1 before mitosis prevents mitotic entry (Lau et al, 2021). Supporting this, the small-molecule inhibitor of Cdk1, RO-3306, effectively arrests cells in G2 phase, as observed through flow cytometry (Vassilev et al, 2006; Tanenbaum et al, 2015; Johnson et al, 2021). We tested various concentrations of RO-3306 in RPE1 cells to analyze the specific cell cycle stages arrested. Cells were incubated with 1, 3, 6, or 10 µM of RO-3306 for 24 h, fixed, and then the cell cycle stages were determined using an immunofluorescence-based cell cycle identification method. We found that treatment with 3 and 6 µM RO-3306 efficiently accumulated cells in the G2 phase, with 60% and 59% of cells, respectively, whereas only 12–13% of cells accumulated in G2 at 1 and 10 µM (Fig 3A–C). Surprisingly, most cells treated with 10 µM RO-3306 were arrested in the G1 phase (Fig 3C), indicating that a high concentration of RO-3306 may inhibit other Cdks in addition to its primary target, Cdk1 (Jorda et al, 2018). In RPE1 cells, 1 µM of RO-3306 was insufficient to arrest cells in the G2 phase (Fig 3C). Treatment with 3 µM RO-3306 resulted in nearly equal populations of early and late G2 phase cells (28% and 32%, respectively), whereas 6 µM RO-3306 predominantly arrested cells in early G2 phase, highlighting Cdk1 activity plays an important role for the progression from early to late G2 phase (Fig 3C). Notably, we observed a subset of interphase cells exhibiting bubbled nuclei specifically in the 3 µM RO-3306-treated groups, likely because of the partial inhibition of Cdk1 activity (Fig 3D). Next, we examined the mitotic index after RO-3306 washout. We quantified mitotic cells at 2 h post-washout in STLC-contained growth medium. Cells treated with 3 and 6 µM RO-3306 exhibited ~35% mitotic cells at 2 h post-washout, suggesting that under these conditions, around 60% of G2-arrested cells successfully progressed into mitosis within this time frame. Interestingly, no mitotic cells were observed after washout in cells treated with 10 µM RO-3306 (Fig 3E), indicating that cells cannot efficiently recover at this concentration.

To further validate our quantification results obtained in fixed-cell analysis, we performed live-cell imaging using RPE1 cells

---

aphidicolin washout (5 µg/ml). From left to right, n = 350, 375 (from two replicates). **(F)** Schematic timeline of live-cell imaging sequence (top). Representative live-cell imaging of H2B-GFP expressing RPE1 cells treated with either DMSO (control) or aphidicolin (2.5 or 5 µg/ml for 24 h) (bottom). Aphidicolin was washed out before imaging. Mitotic cells are indicated with magenta arrows. Imaging was performed at least two independent replicates.

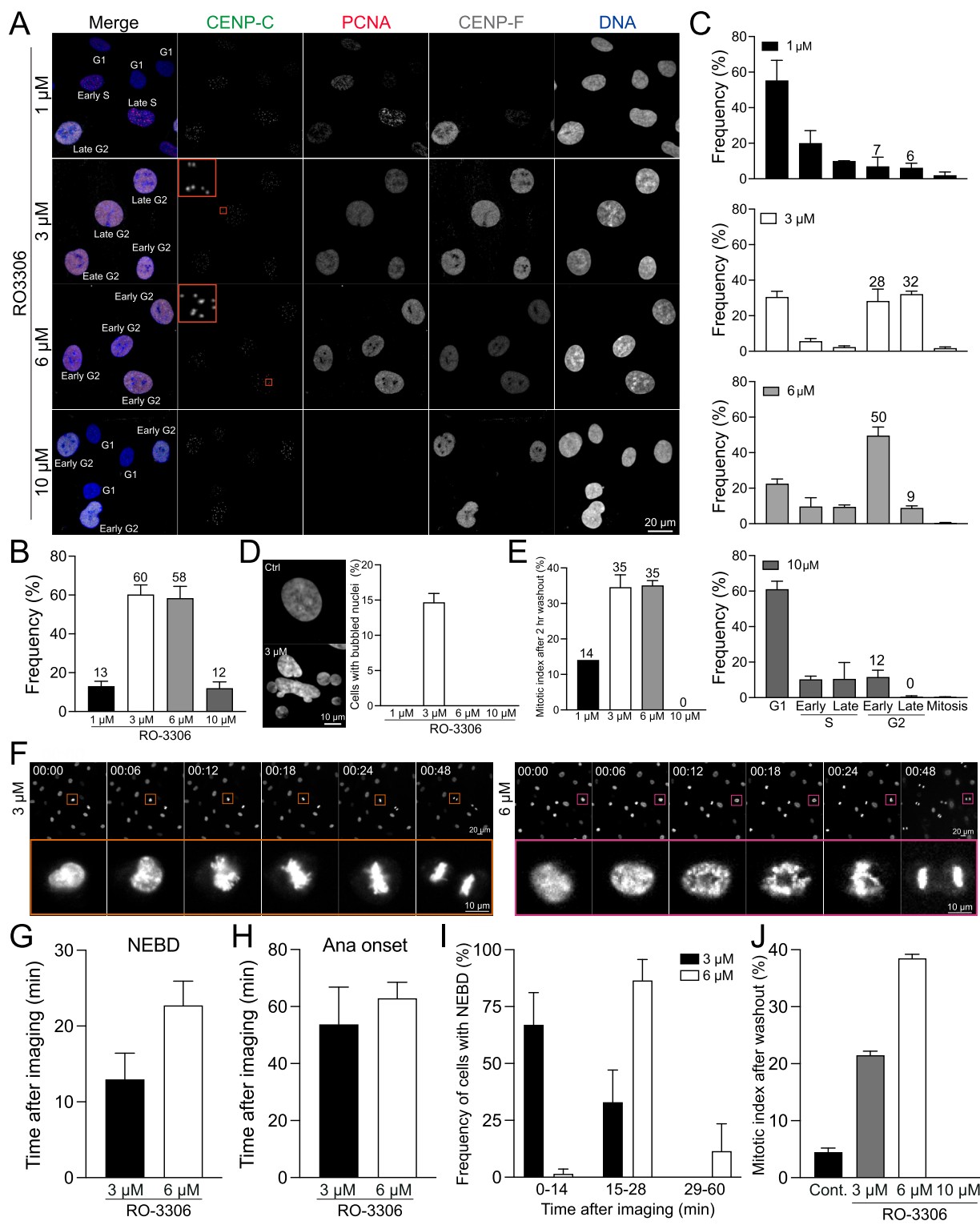

**Figure 3.  G2 phase synchronization and release in RPE1 cells using RO-3306.**
**(A)** Representative immunofluorescence images of RPE1 cells under control conditions or treated with RO-3306 (1, 3, 6, or 10 μM for 24 h), stained with antibodies for CENP-C, PCNA, and CENP-F. **(A, B)** Percentage of cells in G2 phase in condition (A). From left to right, n = 404, 362, 318, 388 (from two replicates). **(A, C)** Proportion of cells in each stage of cell cycle in condition (A). **(D)** Representative DNA images and percentage of cells with bubbled nucleus. From left to right, n = 404, 362, 318, 388 (from two replicates). **(E)** Mitotic index at 2 h after RO-3306 washout in growth media containing STLC. From left to right, n = 443, 641, 513, 430 (from two replicates). **(F)** Representative live-cell imaging of Nuc650 labeled RPE1 cells treated with either 3 or 6 μM of RO-3306 24 h. RO-3306 was washed out before imaging. **(G)** Average time to nuclear envelope breakdown post-imaging initiation, in cells treated with RO-3306 at concentrations of 3 or 6 μM for 24 h. The RO-3306 treatment was washed out before imaging commenced. n = 60 and 60 (from left to right, two replicates). **(G, H)** Average time to anaphase onset in cells from condition (G). n = 60 and 60 (from left to right,

immediately after treatment of 3 or 6 µM RO-3306 (Fig S6A and B). Whereas control cells consistently exhibited mitotic progression during live-cell imaging, cells treated with 6 µM RO-3306 did not show any progress to mitosis, indicating that 6 µM of RO-3306 effectively inhibits mitotic entry. Although mitotic index was significantly reduced in cells treated with 3 µM RO-3306, the subset of cells that entered mitosis experienced a slight but significant delay in mitotic duration and nuclear bubbling (Fig S6A arrow), consistent with observations in fixed-cell analysis (Fig 3D). These results demonstrate that RO-3306 at concentrations of 3 and 6 µM effectively arrests most of the cells in the G2 phase, whereas concentrations of 1 and 10 µM fail to exhibit this effect. Importantly, a subset of G2-arrested cells progresses into mitosis in 3 µM RO-3306 treated cells. These mitotic cells displayed significant errors in both mitotic progression and anaphase, resulting in nuclear bubbling (Fig S6B and C) (Voets et al, 2015).

We assessed the efficacy of G2 phase synchronization using additional Cdk1 inhibitors, flavopiridol and BMS-265246 (Fig S7A–E) (Dickson & Schwartz, 2009; Kang et al, 2020; Li et al, 2024). Among the inhibitors tested, RO-3306 demonstrated the highest potency, achieving ~60% of cells in the G2 phase at concentrations of 3~6 µM in RPE1 cells. In contrast, flavopiridol (2–10 µM) and BMS-265246 (5 µM) arrested approximately 10–20% and ~40% of cells in the G2 phase, respectively (Fig S7B and E). Notably, 48 h of BMS-265246 treatment did not increase the proportion of G2-phase cells (25%) compared with 24 h (41%) (Fig S7B). Instead, it led to a significant increase in the proportion of G1 phase cells (from 43% to 71%), indicating that both flavopiridol and BMS-265246, although less effective at inducing G2 arrest, also arrested cells in G1 phase at higher concentrations, similar to RO-3306 treatment at 10 µM. Consistent with RO-3306 treatment, cells treated with BMS-265246 also exhibited nuclear bubbling (Fig S7C).

Next, we examined recovery after RO-3306 washout using live-cell imaging (Fig S7F). Cells treated with 3 and 6 µM RO-3306 underwent the nuclear envelope breakdown (NEBD) within 20 min and progressed to anaphase onset ~60 min after RO-3306 washout. In contrast, no mitotic cells were observed after the washout of 10 µM RO-3306 (Figs 3E, G, and H and S7F). After washout, cells treated with 3 µM RO-3306 entered mitosis significantly faster than those treated with 6 µM (Fig 3F–I). However, a subset of these cells displayed mitotic errors and bubbled nuclei even after washout (Fig S6D). Additional live-cell imaging experiments with RO-3306 washout were conducted using STLC-supplemented media, and the mitotic index was measured 2 h after the washout. Consistent with the fixed-cell analysis, ~22~38% of the cells were found to be arrested in mitosis in 3 and 6 µM RO-3306 treated cells, compared with only 4% in the control group (Figs 3J and S6E). Collectively, RO-3306 at concentrations between 3 to 6 µM effectively accumulate cells in G2 phase; however, subsets of cells treated with 3 µM, with or without washout, exhibited mitotic errors and nuclear bubbling. Higher concentration of RO-3306 (10 µM in RPE1 cells) fails to synchronize RPE1 cells in G2 phase and prevent, at least efficient,

recovery to a normal cell cycle progression even after RO-3306 removal.

## Cell cycle synchronization in prometaphase

Microtubule depolymerizers, including nocodazole and colcemid, have traditionally been used for mitotic synchronization because of their ability to effectively disrupt spindle formation and prevent chromosome segregation (Florian & Mitchison, 2016; Surani et al, 2021; Hadfield et al, 2022). However, despite their reversible nature, cells treated with these drugs and subsequently washed exhibit a marked increase in severe mitotic defects because of the lack of microtubule dynamicity (Cavazza et al, 2016; Worrall et al, 2018). Because of these limitations, our study used STLC and monastrol, potent Eg5 inhibitors, as alternative agents to arrest cells in mitosis (Kapoor et al, 2000; Florian & Mitchison, 2016; Hadfield et al, 2022). After NEBD, chromosomes undergo dynamic interactions with microtubules during prometaphase, including the capture of kinetochores and the establishment of bipolar spindles required for metaphase plate formation. Whereas high concentrations of traditional microtubule depolymerizers obliterate microtubules, Eg5 inhibitors do not prevent microtubule assembly at kinetochores. Instead, it impedes centrosome separation necessary for bipolar spindle formation, resulting in prometaphase arrest when maintaining partial kinetochore-microtubule interactions (Skoufias et al, 2006; Chen et al, 2024a Preprint). Consequently, removing Eg5 inhibitors is thought to facilitate a more effective recovery than treatment with microtubule depolymerizers (Bakhoum et al, 2009).

In our study, we treated cells with 2, 5, or 10 µM STLC for 24 h and assessed the mitotic index. The results showed that 5 and 10 µM concentrations achieved ~60% synchronization efficiency, whereas 2 µM STLC treatment exhibited nearly equivalent synchronization efficiency as untreated control (Fig 4A and B). As expected, in the presence of 5 and 10 µM STLC, almost 100% of the mitotic cells were arrested in prometaphase and exhibited monopolar spindles (Fig 4C and D). Similar to STLC, treatment with both 2 and 4 µM monastrol for 24 h arrested ~60% of cells in prometaphase (Fig S8A and B). Notably, whereas monastrol achieves comparable levels of cell arrest at slightly lower concentrations than STLC, it is significantly more expensive. These results confirm the efficiency of > 5 µM STLC and > 2 µM of monastrol in synchronizing cells at prometaphase. For applications requiring a higher purity of prometaphase populations, we recommend using a mitotic shake-off technique (Zwanenburg, 1983) after STLC synchronization, which yielded nearly 100% pure prometaphase population (Fig 4A and E). We validated the immunofluorescence-based quantification of STLC synchronization by live-cell imaging. RPE1 cells treated with 5 or 10 µM STLC demonstrated a gradual and efficient accumulation in prometaphase, with ~80% of cells arrested in this stage after 24 h (Fig 4F [arrows], Figs 4G and S8C). Nearly 100% of these prometaphase cells formed monopolar spindles because of Eg5

---

two replicates). **(G, I)** The proportion of cells that enter nuclear envelope breakdown after the start of imaging for the same treatments of (G). n = 60 and 60 (from left to right, two replicates). **(J)** Mitotic index measured 2 h after washout of 0 (control), 3, 6, or 10 µM RO-3306 in STLC-supplemented media. From left to right, n = 482, 511, 480, 653 (from two replicates).

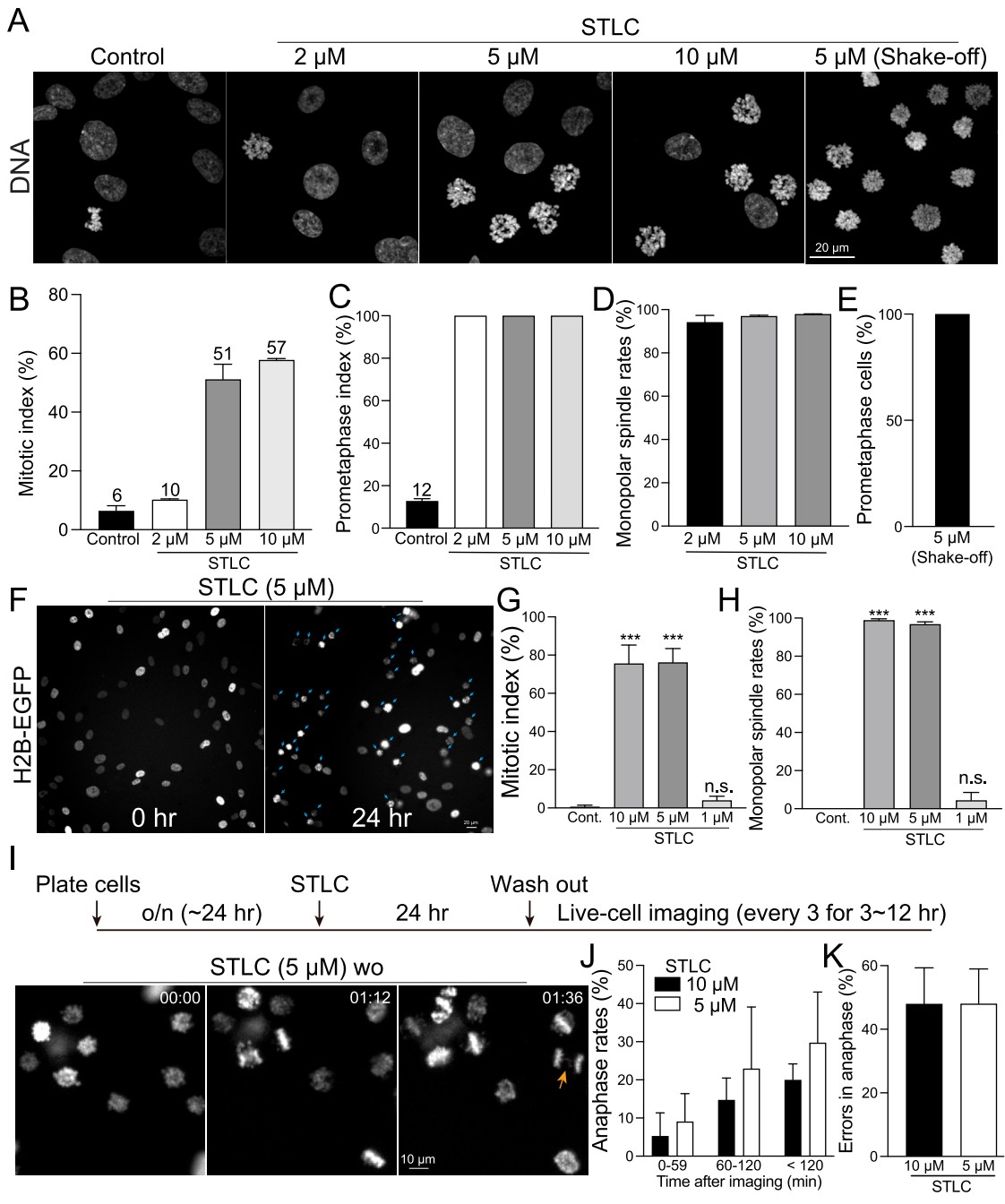

**Figure 4. Prometaphase synchronization and release in RPE1 cells using STLC.**
**(A)** Representative confocal images of DNA in RPE1 cells under control conditions, treated with STLC at concentrations of 2, 5, or 10 $\mu$M, and post-mitotic shake-off after treatment with 5 $\mu$M STLC. **(B)** Mitotic index of cells under control condition compared to those treated with STLC (2, 5, or 10 $\mu$M). From left to right, n = 424, 452, 406, 789 (two replicates). **(B, C)** Prometaphase index corresponding to the treatments described in (B). From left to right, n = 341, 261, 208, 81 (two replicates). **(D)** Percentage of mitotic cells displaying a monopolar spindle after treatment with STLC at 2, 5, or 10 $\mu$M. From left to right, n = 193, 204, 65 (two replicates). **(E)** Mitotic index after mitotic shake-off in cells treated with 5 $\mu$M STLC. n = 503 (two replicates). **(F)** Representative live-cell imaging of H2B-GFP-expressing RPE1 cells treated with 5 $\mu$M STLC. Blue arrows indicate mitotic cells. **(G)** Mitotic index in live-cell imaging under control condition and after treatment with STLC at concentrations of 1, 5, or 10 $\mu$M for 24 h. **(G, H)** Proportion of mitotic cells with a monopolar spindle after the treatments outlined in (G). **(I)** Schematic timeline of live-cell imaging sequence (top). Representative live-cell imaging of H2B-GFP expressing RPE1 cells treated with either DMSO (control) or 5 $\mu$M STLC for 24 h, after which STLC was washed out (bottom). A mitotic cell with lagging chromosomes is highlighted with an orange arrow. **(J)** Proportion of cells progressing to anaphase. **(K)** Percentage of anaphase cells exhibiting errors, including lagging chromosomes and chromosome bridges. n = 679 and 518 (from left to right (J, K), two replicates).

inhibition (Fig 4H). In contrast, most cells treated with 1 $\mu$M of STLC could proceed through division (Figs 4G and S8C). Importantly, there was no significant increase in apoptotic cell death among cells treated with any concentration of STLC compared with the control during 24 h of live imaging (Fig S8C). These observations are in alignment with the results obtained from immunofluorescence-

based quantifications, which showed that treatment with 5 and 10 μM of STLC effectively arrests cells in prometaphase.

We next investigated whether mitotic cells arrested by STLC could exit mitosis after washout. For this experiment, RPE1 cells were incubated with STLC at concentrations of 5 or 10 μM for 24 h. After the washout, we immediately commenced high-temporal-resolution live-cell imaging (Figs 4I and S8D). We quantified the percentage of arrested cells that entered anaphase within 2 h post-washout. Our results showed that ~20% and 30% of the cells arrested in prometaphase progressed to anaphase within 2 h after washout of 5 or 10 μM STLC, respectively (Fig 4J). Notably, only 10% of cells underwent anaphase within the first hour. Among these divided cells, about 50% exhibited errors during anaphase (Fig 4I [arrow], Figs 4K and S8D [arrow]). These findings indicate that only a subset of STLC-arrested cells is able to enter anaphase immediately after the washout.

### Cell cycle synchronization in metaphase, anaphase, and telophase

The transition from metaphase to anaphase necessitates the degradation of Cyclin B and securin (Han & Li, 2014). This degradation activates separase, allowing it to cleave the cohesion between sister chromatids and enabling their segregation. Consequently, proteasome inhibitors such as MG132 have been identified to effectively induce metaphase arrest (Santamaria et al, 2007; Daum et al, 2011; Tipton & Gorbsky, 2022). Previous studies have demonstrated that cells treated with MG132 maintain the metaphase plates, resulting in kinetochores experiencing heightened tension compared with those in normal metaphase (Wan et al, 2009). This increased tension is evidenced by the observed increases in the intra- and inter-kinetochore stretch. However, it is important to note that proteasome inhibitors lack specificity in mitotic processes, raising concerns about their potential to disrupt various cell cycle regulations inadvertently. To support this, unlike STLC, RPE1 cells treated with 10 μM MG132 for 24 h did not show a significant increase in mitotic index (Fig 5A and B). On the other hand, metaphase cells exposed to long-term MG132 treatment exhibited significant defects in chromosome alignment (Fig 5A and C), likely because of cohesion fatigues (Daum et al, 2011). To further validate this observation, we performed live-cell imaging on cells treated with 10 μM MG132 (Fig 5D). Although these cells established and maintained a metaphase plate for ~2 h after NEBD, the spatial organization of chromosomes became disorganized thereafter, leading to misaligned chromosomes and apoptotic cell death. These results demonstrate that using MG132 alone is insufficient for synchronizing cells in metaphase, anaphase, and telophase. To enrich populations of metaphase cells, we used a combination approach involving RO-3306 for G2 cell synchronization followed by MG132 treatment (Figs 5E and S9A). As most cells arrested by RO-3306 progress to NEBD within 1–2 h, we investigated the effects of MG132 treatments for 1 or 2 h on the synchronization efficacy of metaphase cells after RO-3306 washout. Our findings reveal that the combination of RO-3306 and MG132 effectively increases the population of metaphase cells (Figs 5E and S9A). Interestingly, ~40–50% of cells arrested in metaphase after 2 h of MG132 treatment fail to initiate anaphase within 2 h after MG132 washout (Fig

5F). In contrast, nearly 100% of these cells subjected to 1-h MG132 treatment enter anaphase. This phenotype is not rescued by reducing the concentration of MG132 to 5 μM, suggesting that MG132 treatment exceeding 1 h or arresting cells in metaphase for longer than 1 h impedes anaphase entry even after washout.

For anaphase cell synchronization, cells treated with 5 μM of MG132 for 1 h exhibited anaphase onset immediately after washout, with ~80% of cells entering anaphase within 30 min after MG132 removal (Figs 5G and S9B). Conversely, cells treated with 10 μM of MG132 showed ~60% of cells entering anaphase within a range of 30–60 min after washout. The telophase population peaked between 30 and 60 min in cells treated with 5 μM MG132 and between 45 and 75 min in cells treated with 10 μM MG132 after washout (Figs 5H and S9C). About 50% of anaphase cells exhibited errors in both 5 and 10 μM MG132-treated cells for 2 h, whereas ~16–30% of these cells exhibited errors after 1 h of treatment (Fig S9D). Although no metaphase-arrested cells treated with 5 μM MG132 for 1 or 2 h exhibited apoptotic cell death within 2 h after washout, 2–5% of cells exhibited apoptotic cell death in cells treated with 10 μM MG132 for 1 and 2 h, respectively (Fig S9E). In addition, no anaphase cells were found in cells treated with 10 μM MG132 at the beginning of imaging, whereas cells treated with 5 μM MG132 for 1 h occasionally entered anaphase upon imaging (Fig S9F). Collectively, the combination of RO-3306 G2 cell cycle synchronization and a 1-h treatment with MG132 at concentrations ranging from 5 to 10 μM is capable of accumulating cells in healthy metaphase. Depending on the desired accumulation of anaphase and telophase cells, either 5 or 10 μM MG132-treated cells can be used, tailored to the specific timing requirements of subsequent experiments. Whereas 5 μM MG132-treated cells exhibit a higher rate of proper anaphase progression compared with those treated with 10 μM MG132 upon washout, these cells promptly progress into anaphase upon removal of MG132. On the other hand, 10 μM MG132-treated cells offer slightly more time for the preparation of subsequent procedures.

### Limitation of this study

For our synchronization method, we optimized the protocol using the RPE1 cell line, a normal, non-transformed human cell line expressing WT p53 (Bowden et al, 2020). It has been reported that certain inhibitors, particularly Cdk inhibitors, exhibit varying efficacies across different cell lines (Trotter & Hagan, 2020; Johnson et al, 2021). This variability may be attributed to the differential activities of Cdks in distinct cell types. A study demonstrated that in cancer cells, Cdk2 can compensate for the loss of Cdk1 during mitotic entry when Cdk1 is rapidly degraded using the auxin-degron system (Lau et al, 2021). However, this compensation does not occur in normal cells. Whereas our optimized inhibitor concentrations serve as a valuable reference for cell cycle synchronization, adjustments may be necessary when applied to other cell lines, especially those with p53 deletions or mutations. To efficiently synchronize most cells at specific cell cycle stages with minimal process, we incubated cells with most inhibitors for 24 h in this study. However, this prolonged incubation may induce artificial stress responses or subtle, undetectable defects in our assay, after the washout. Furthermore, our investigation into drug recovery focused on the short-term effects after drug release, with the

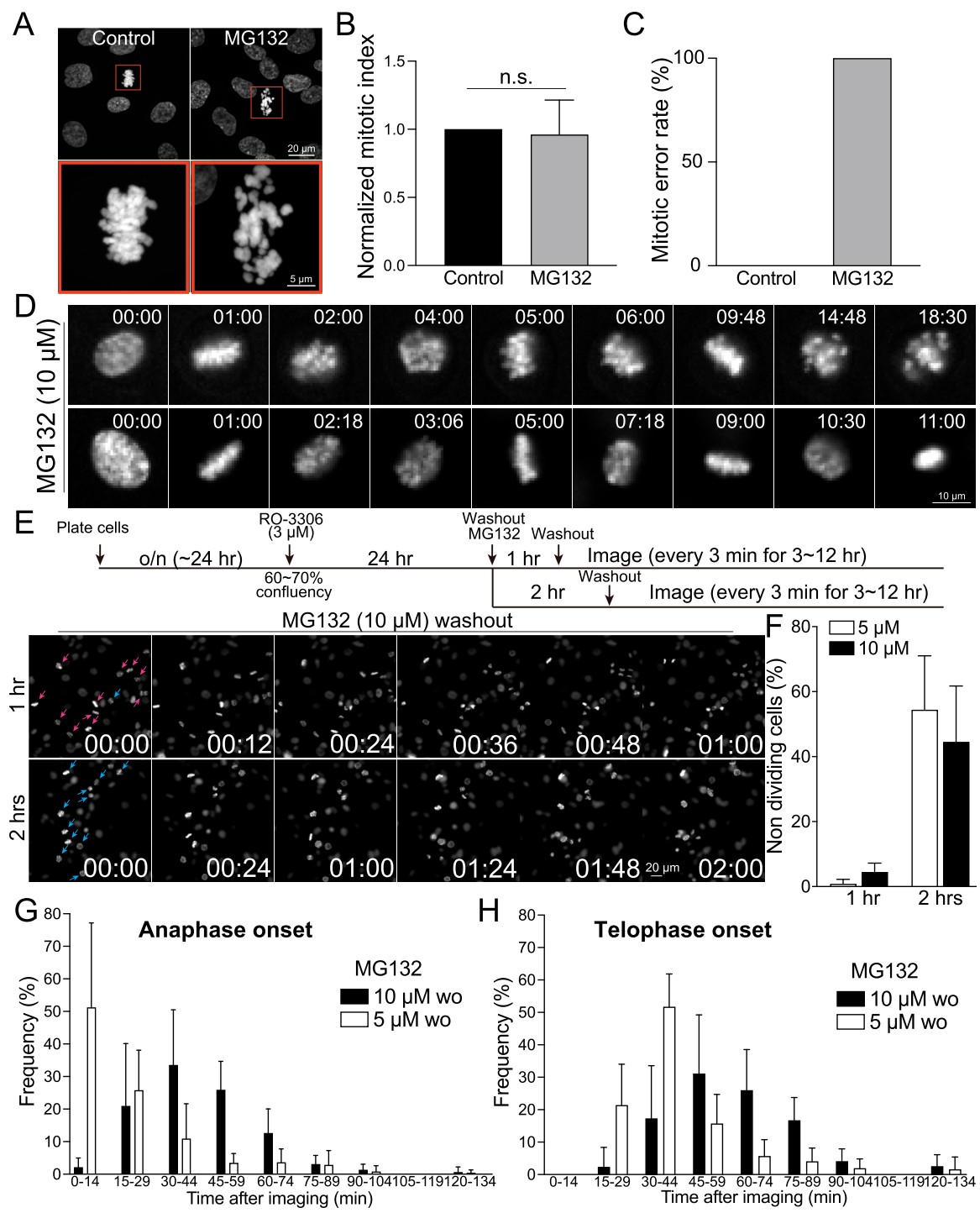

**Figure 5.  Metaphase, anaphase, and telophase synchronization using both RO-3306 and MG132.**
**(A)** Representative confocal images of DNA in RPE1 cells under control conditions or treated with MG132 (10 µM) for 24 h. **(B)** Mitotic index of cells under control conditions compared with those treated with MG132 (10 µM) for 24 h. From left to right, n = 619, 647 (two replicates). **(C)** Mitotic error rates in control or cells treated with MG132 (10 µM) for 24 h. Left: n = 12 (from 619 cells from two replicates), Right: n = 12 (from 647 cells from two replicates). **(D)** Representative live-cell imaging of H2B-GFP-expressing RPE1 cells treated with 10 µM MG132. **(E)** Schematic timeline of live-cell imaging sequence (top). Representative live-cell imaging of H2B-GFP expressing RPE1 cells treated with 10 µM MG132 for either 1 or 2 h, after which MG132 was washed out (bottom). Before the treatment of MG132, cells were incubated with RO-3306 for 24 h. **(E, F)** Proportion of non-dividing mitotic cells after the treatments outlined in (E). n = 196, 237, 257, 164 (from left to right, two replicates). **(E, G, H)** Proportion of cells entering anaphase onset or telophase onset in condition (E). n = 237, 196 (from two replicates).

possibility of uncovering additional defects in subsequent cell cycle phases.

# Discussion

Cell cycle synchronization is a commonly used method to accumulate cell populations in specific stages of the cell cycle to study stage-specific mechanisms and regulations. To achieve this, treatments with inhibitors targeting cell cycle-specific and essential kinases or proteins are commonly used (Dickson & Schwartz, 2009; Mills et al, 2017). However, these inhibitors often induce irreversible effects at higher concentrations and demonstrate inefficacy at lower concentrations. By integrating our immunofluorescence-based cell cycle identification method with cell synchronization and subsequent washout protocols, we carefully evaluated the efficacy of various inhibitors used for cell cycle synchronization. Our findings reveal that all tested inhibitors induced certain defects and resulted in irreversible arrest in the cell division cycle (Figs 1, 2, 3, 4, and 5). For example, RPE1 cells synchronized in the G1 phase using optimal concentrations of palbociclib exhibited ~30% arrested cells in the G1 phase 18 h after washout (Fig 1C). Similarly, cells synchronized to the S phase by aphidicolin also exhibited ~20% cells in the G1 phase 6 h after washout (Fig 2B and C). Surprisingly, RO-3306 is now more frequently used for G2 synchronization. Higher than optimal concentrations showed no G2 phase synchronization (Fig 3C), indicating that high concentrations of RO-3306 might inhibit other Cdks, although RO-3306 is considered a selective inhibitor for Cdk1 (Jorda et al, 2018). Cells treated with the optimal concentration of RO-3306 can significantly accumulate in the G2 phase (~60%); however, only ~60% of these G2 phase cells can immediately enter mitosis after washout (Fig 3E). Treatment with MG132 for more than 1 h causes irreversible defects in metaphase cells, both with and without washout (Fig 5). We summarize our recommended conditions for cell synchronizations at each stage of the cell cycle in RPE1 cells in Table S1.

We demonstrated that all the inhibitors we tested were unable to prevent some irreversible effects or other defects. This may be because of off-target effects of the inhibitors or difficulties in achieving complete washout. To circumvent these issues, developing conditional knockout cell lines for cell cycle kinases could be a viable alternative, although it requires additional effort to generate these strains. Notably, a previous study demonstrated that rapid depletion of Cdk1 in HeLa cells still allowed entry into mitosis, as Cdk2 compensates for Cdk1's role in mitotic entry but not mitotic exit (Lau et al, 2021). Interestingly, RO-3306 effectively arrested HeLa cells in the G2 phase (Vassilev et al, 2006). This might be because RO-3306 inhibits not only Cdk1 but also other Cdks. This suggests that the use of inhibitors can effectively arrest cells at a specific point in the cell cycle, overcoming potential compensatory effects by other kinases. This approach may be more effective than using conditional knockout cell lines for targeting cell cycle kinases in certain cell types. Nevertheless, our detailed analysis of cell cycle inhibitors and the optimization of reversible and effective cell synchronization in RPE1 cells will provide a standard and serve as a reference for future research.

# Materials and Methods

## Cell culture

Human RPE1 cells were originally obtained from the America n Type Culture Collection (ATCC). RPE1 H2B-EGFP cells were obtained from Dr. Beth Weaver. RPE1 and RPE1 H2B-EGFP cells were grown in DMEM high glucose (Cytiva Hyclone; SH 30243.01) supplemented with 1% penicillin–streptomycin, 1% L-glutamine, and 10% FBS under 5% $CO_2$ at 37°C in an incubator.

## Cell synchronization

Cells were plated one day before inhibitor treatment, reaching 60–70% confluency at the time of treatment, except for the double thymidine block, where cells were at ~30% confluency before the first thymidine treatment. Inhibitors used for cell cycle synchronization included palbociclib, thymidine, ribociclib, abemaciclib, aphidicolin, hydroxyurea (HU), RO-3306, BMS-265246, flavopiridol, STLC, monastrol, and MG132, detailed in Table S1. The cell density at the time of treatment was critical for successful synchronization. Briefly, cells were synchronized at the G1 phase by incubating with palbociclib, ribociclib, or abemaciclib for 24 h. The double thymidine block was performed as follows: cells were treated with 2 mM thymidine for 16 h, followed by a 9-h washout, and then subjected to a second 16-h thymidine treatment. To induce the G0 phase, cells were cultured in serum-free medium for 48 h. If re-entry into the cell cycle was required, the medium was supplemented with 20% FBS to stimulate cell cycle progression. For S phase synchronization, cells were treated with aphidicolin or HU for 24 h. G2 phase synchronization involved a 24-h incubation with RO-3306, BMS-265246, or flavopiridol. For synchronization at metaphase, anaphase, and telophase, cells were treated with MG132 for 1 h after a 24-h RO-3306 treatment. Washout experiments were generally performed in media containing STLC to prevent re-entry into the subsequent cell division cycle.

## Live-cell imaging

RPE1 or RPE1 H2B-EGFP cells were plated on 4-chamber 35 mm glass bottom dishes (four chamber with #1.5 glass; Cellvis) or μ-slide eight well high glass bottom (80807; ibidi) at least one day before imaging. After 24 h of plating, cells were treated with inhibitors for cell synchronization (see the Cell synchronization section) and, if necessary, subjected to washout before commencing live-cell imaging. For RPE1 cells, NucSpot 650 (Cat# 40082; Biotium) was used to label the cells 2 h before live-cell imaging, following the manufacturer's protocol. Live-cell imaging was performed using a Nikon Ti2 inverted microscope equipped with a Hamamatsu Flash v2 camera, spectra-X LED light source (Lumencor), Shiraito PureBox with a STXG stage-top incubator (TokaiHit), and a Plan Apo 20x objective (NA = 0.75) controlled by Nikon Elements software. Cells were recorded at 37°C with 5% CO2 in a stage-top incubator using the feedback control function to accurately maintain temperature of growth medium (Tokai Hit,

STXG model). For non-wash out conditions, images were recorded for ~24 h at 30 min intervals with three z-stack images acquired at steps of 3 μm for each time point. For washout experiments, most of images were recorded for 3–24 h at 3 or 6 min intervals with or without STLC-supplemented media.

### Immunofluorescence (IF)

Accurate identification of cell cycle stages was achieved using ImmunoCellCycle-ID, a tool we recently developed (Chen et al, 2024b). The following primary and secondary antibodies, along with a DNA dye, were used: anti-CENP-F (kindly gifted by Dr. Stephen Taylor), PCNA (sc-56; Santacruz), CENP-C (PD-030; MBL), Acetylated Tubulin (T7451; Sigma-Aldrich), BRCA1 (sc-6954; Santacruz), phospho-Rb (8516; Cell Signaling), DAPI (D9542; Sigma-Aldrich), Guinea Pig IgG-Alexa 647 (706-606-148; JacksonImmuno), Sheep IgG-Rhodamine Red X (713-546-147; JacksonImmuno), and Mouse IgG (715-546-150; JacksonImmuno). RPE1 cells were fixed by 4% PFA (Sigma-Aldrich) for experiments in Figs 3E, 4A–E, and 5A–C and S8A and B, whereas all other IF experiments were fixed using 100% ice-cold Methanol. Cells which fixed with PFA were then permeabilized by 0.5% NP40 (Sigma-Aldrich) and incubated with 0.1% BSA (Sigma-Aldrich). Stained samples were imaged by CSU W1 SoRa spinning disc confocal, which was equipped with Uniformizer and a Nikon Ti2 inverted microscope with a Hamamatsu Flash V2 camera and a 60x Oil objective (NA = 1.40). Microscope system was controlled by Nikon Elements software (Nikon).

### Cell cycle stage identification

Accurate identification of cell cycle stages was achieved using the ImmunoCellCycle-ID method, as detailed in a previous study (Chen et al, 2024b). Briefly, G1 phase cells exhibited uniform nuclear PCNA signals without nuclear CENP-F staining. S phase cells were characterized by distinct punctate nuclear PCNA patterns accompanied by nuclear CENP-F signals. G2 phase cells displayed uniform nuclear PCNA signals, high nuclear CENP-F expression, and prominent kinetochore signals of CENP-C, including paired configurations. G0 phase cells were distinguished by the presence of unique cilium formation and the absence of punctate nuclear PCNA signals. Mitotic cells were identified based on distinct nuclear morphology. In aphidicolin-treated cells, additional markers such as BRCA1 and pRb were used for enhanced analysis. The detailed dynamics of these markers throughout the cell cycle are presented in the Results section.

### Mitotic shake-off

RPE1 cells were treated with 5 μM of STLC for 24 h, after which mitotic cells were collected by shaking. The growth medium was then centrifuged to concentrate the cells. Subsequently, these cells were cytospin onto coverslips, fixed with 4% PFA, and stained with DAPI (refer to the Immunofluorescence section for details).

### Image analysis

Image analysis was performed using Nikon Elements software (Nikon) or Metamorph (Molecular Devices).

### Statistics

All experiments were independently repeated two to three times for mitotic duration measurements. P-values were calculated using one-way ANOVA. P-values < 0.05 were considered significant.

## Data Availability

All data are included in the manuscript and/or Supplemental Material. Original images used in this study and additional images related to this study are available from the corresponding author (A Suzuki) upon reasonable request.

## Supplementary Information

## Acknowledgements

We would like to thank Yu-Chia Chen, Yuhi Hara, Takanori Tsuchiya, Yoshitaka Sekizawa, Yokogawa Electric Corporation, and Tokai Hit for valuable suggestions, critical equipment and technical support. Part of this work is supported by Wisconsin Partnership Program, Research Forward from the Office of the Vice Chancellor for Research (OVCR) Wisconsin Alumni Research, start-up funding from University of Wisconsin-Madison SMPH, UW Carbone Cancer Center, and McArdle Laboratory for Cancer Research, NIH grant R35GM147525 and CRNA grant U54AI170660 (to A Suzuki).

### Author Contributions

Y-L Chen: data curation, formal analysis, validation, investigation, visualization, methodology, and writing—review and editing.
S Reddy: data curation and writing—review and editing.
A Suzuki: conceptualization, resources, data curation, supervision, funding acquisition, investigation, visualization, methodology, project administration, and writing—original draft, review, and editing.

### Conflict of Interest Statement

The authors declare that they have no conflict of interest.

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
