## [Reviewer comments · Life Science Alliance]

Life Science Alliance

Reversible and effective cell cycle synchronization method for studying stage-specific processes

Yu-Lin Chen, Syon Reddy, and Aussie Suzuki

DOI: <https://doi.org/10.26508/lsa.202403000>

Corresponding author(s): Aussie Suzuki, University of Wisconsin-Madison

Review Timeline:

Submission Date:	2024-08-19
Editorial Decision:	2024-09-26
Revision Received:	2025-01-19
Editorial Decision:	2025-02-10
Revision Received:	2025-02-20
Accepted:	2025-02-21

Transaction Report:

September 26, 2024

Re: Life Science Alliance manuscript #LSA-2024-03000

Dr. Aussie Suzuki
University of Wisconsin-Madison
1111, Highland Ave, WIMR, Room 6533, UW-Madison
Madison, WI 53705

Dear Dr. Suzuki,

Thank you for submitting your manuscript entitled "Reversible and effective cell cycle synchronization method for studying stage-specific investigation" to Life Science Alliance. The manuscript was assessed by expert reviewers, whose comments are appended to this letter. We invite you to submit a revised manuscript addressing the Reviewer comments.

Thank you for this interesting contribution to Life Science Alliance. We are looking forward to receiving your revised manuscript.

Sincerely,

B. MANUSCRIPT ORGANIZATION AND FORMATTING:

Reviewer #1 (Comments to the Authors (Required)):

The manuscript described the characterization of the multiple cell cycle phases synchronization of RPE1. The presented methodology, data, and results are straightforward and easily comprehensible, making them a valuable resource for the research community. Unfortunately, the study is significantly constrained by the dependence solely on inhibitor concentration as the single parameter for synchronization. Further investigation is required to address the following concerns in the study:

1. Inhibitor selection: While palbociclib is the most often used CDK4/6 inhibitor, there are many alternative options available (such as Abemaciclib and ribociclib). While all of the CDK4/6 inhibitors target CDK4/6, their off-target inhibition could differ, resulting in potential differences in their efficiency for the purpose of cell cycle synchronization. The exact same issues also extend to the DNA polymerase, Eg5, and proteasome inhibitors.

2. The inhibitor-based synchronization strategies in comparison to traditional synchronization approaches like thymidine and serum starvation should be discussed.

Reviewer #2 (Comments to the Authors (Required)):

This is overall a study that is well done and the table does a good job summarizing strategies to use for cell synchronization. There is a small number of suggestions to improve the paper. This would just need some further data mining from the existing data sets (live imaging).

The Palbociclib release experiment (Fig 1c) seems to be showing that resumption of cell cycle at all concentrations is not so good. I would expect most of the cells to reach mitosis and become arrested during this 18 hr release. Or, are cells slipping out of mitosis. The live imaging of this washout experiment could be quantified in terms of mitotic index. That would reveal the kinetics and if cells are slipping out of mitosis.

The live imaging in the aphidicolin experiment (Fig 2d) could be quantified in terms of cumulative mitotic index to reveal the kinetics of cells arriving at mitosis. That would be valuable.

In the RO washout experiment (Fig 3d) only about half the G2 cells entered mitosis within 2 hrs. This seems quite poor recovery. In Fig 3e, 2 hrs after washing out the 6uM RO into STLC medium, only 8% of cells were in mitosis. The shows lack of recovery. However, in Fig 3i, there seem to be about 70% cells in mitosis in the 15-29 minute time points. What is the explanation for this difference?

Minor comments.

1. The title is a bit vague. How about something like, "Reversible and effective synchronization methods for studying stage-specific cell cycle processes."

Reviewer #3 (Comments to the Authors (Required)):

In this manuscript, the authors used several small molecule inhibitors to synchronize human nontransformed RPE1 cells in various phases of the cell cycle. Not surprisingly, they observed that treatment with CDK4/6 inhibitor synchronized cells in G1 phase, aphidicolin at G1/S transition, CDK1 inhibitor in G2 and Eg5 inhibitor in prometaphase. These phenotypes correspond well to the known modes of action of all these molecules. Also not surprisingly, they observed that lower concentrations were usually not as efficient as optimal concentrations of the drugs. On the other hand, too high concentrations showed suboptimal synchronization and/or extended recovery times probably due to the lower specificity towards the expected targets. I am somewhat confused how to evaluate this study as all the observed effects have been reported previously although maybe not in such systematic way. These protocols may be handy for researchers outside from the cell cycle field but in my opinion, they do not bring any new view of the synchronization of the cells. The doses of the compounds were optimized specifically for RPE1 cells, which are widely used in the field. Nevertheless the values are not universally valid for other cell types so similar optimization procedures will be necessary for other cellular models. In the summary, I appreciate the effort of authors with careful optimization of their experimental system but I feel that online protocol might be more suitable form of publication.

Reviewer #4 (Comments to the Authors (Required)):

Asynchronously-proliferating cell populations have cells in different cell cycle phases. Analyzing factors that oscillate in the cell cycle can thus be challenging to interpret. Thus, effective synchronization methods are helpful to address this challenge. In this manuscript, Chen et al., provide a comprehensive guide to synchronization by cell cycle block and release methods for human RPE1 cells using drug inhibitors. They highlight that specific drug concentrations could be effective in synchronizing cells but could also affect normal cell cycle progression post-release. The authors admirably acknowledge that the recommendations they give in this manuscript apply for untransformed RPE1 cells, and may not be applicable for other cell lines especially cancer cell lines, though it is a starting point for extrapolating to other cell lines. This paper will be of interest to readers who intend to use pharmacological inhibitors for RPE1 synchronization and release.

The methods for analyzing cell cycle phase in this study lean heavily on a second study by the same group which is available as a pre-print "ImmunoID...".

We support publication if authors address these major points:

1) The PCNA protein is expressed throughout the cell cycle, and it is not in fact cell cycle-regulated at the level of total protein. The figures as presented give the impression that total PCNA is cell cycle-regulated. On the other hand, PCNA localizes to DNA detectable as foci. Small foci appear in early S phase, and more discrete foci appear in late S phase. It is shown in multiple images that the PCNA signal is not present in G1 cells although it should be if cells were not pre-extracted. Authors mention in the methods section using PFA or methanol for fixation. However, it is not clear which fixation method is used for which protein targets, and whether the signal detected is for soluble proteins or chromatin bound proteins. Based on the data shown here (and in the accompanying pre-print) we assume the PCNA staining was with methanol fixation which would remove soluble protein. Unfortunately, the relevant information is not available in the ImmunoID paper because the supplemental files aren't posted by bioRxiv.

2) Moreover, it is not clearly explained in the text here (or the ImmunoID paper) which proteins are cell cycle regulated or their general mechanism of regulation with citations. For example, CENP-F behaves like an APC/C substrate; is that so? Authors should clarify these points in each figure legend or in text.

3) For synchronizing cells with aphidicolin (Fig. 2), authors argue that aphidicolin effectively inhibits DNA synthesis and arrests RPE1 cells in G1 phase. We disagree that the data support this conclusion, and aphidicolin does not prevent origin firing, just fork progression. In Fig. 2a, small PCNA foci start to appear in the arrested cells. Similarly, CENP-F is shown to be expressed in early S in Fig. 2a, while in suppl Fig. 1, it is depicted as a late S phase marker. We suggest to authors to stain for other G1 or S phase biochemical markers - besides PCNA - to strengthen their claim. For example: CDT1 should be still present if cells are truly arrested in G1 phase without origin firing. If origins have begun firing but forks arrested by aphidicolin, then S phase markers will appear: increase in cyclin A and geminin, loss of p21 and CDT1, etc. PCNA itself is not a sensitive enough measure at this resolution to distinguish very early S from late G1.

4) The authors note that ~30% don't recover from the lower dose of Palbociclib. Most cell lines spontaneously enter quiescence. What fraction of proliferating cells in this study are quiescent?

5) The authors should discuss the potential effects of arrest on subsequent cell behavior that are direct consequences of long-term arrest. Release from an early S phase block will be early S by DNA replication but late S by some molecular markers such as APC/C substrates which accumulated to high levels during the arrest. Stress responses may also contribute to cell cycle progression for some time after release.

Minor points for the authors' consideration:

- Line 59: "reproductive cell cycle" can mean something else other than the cell division cycle.
- Line 102: "presented in this study"
- Line 171: The right-hand side panel of Fig. 2b is not clear. What is the x-axis depicting?
- Line 171: Suppl. Fig 2c: at the 10 ug/mL concentration without washout, cells are depicted as G1 cells although CENP-F (S phase marker) and PCNA is expressed (same concern about bound versus unbound proteins).
- Line 231: Please clarify that 3 μ M and higher concentrations <10 μ M of RO-3306 can efficiently arrest most cells in G2 because the presented data in this paper show that 10 μ M arrest cells in G1 and might inhibit other CDKs, as explained by the authors.
- Line 254: please spell out Nuclear Envelope Breakdown (NBD)
- Line 266: "as expected" instead of "as expect".
- Line 352: Limitations section: It could be helpful to readers to point out that they focused in this paper on short-term recovery post drug washout, and that longer release periods might reveal additional defects in subsequent cell cycles.

Point-by-point response

Reviewer #1 (Comments to the Authors (Required)):

The manuscript described the characterization of the multiple cell cycle phases synchronization of RPE1. The presented methodology, data, and results are straightforward and easily comprehensible, making them a valuable resource for the research community. Unfortunately, the study is significantly constrained by the dependence solely on inhibitor concentration as the single parameter for synchronization. Further investigation is required to address the following concerns in the study:

We appreciate the reviewer's recognition of the significance of our study. In the revised manuscript, we have expanded our analysis to include traditional synchronization methods and alternative inhibitors. The detailed results of these analyses are provided below step-by-step responses.

1. *Inhibitor selection: While palbociclib is the most often used CDK4/6 inhibitor, there are many alternative options available (such as Abemaciclib and ribociclib). While all of the CDK4/6 inhibitors target CDK4/6, their off-target inhibition could differ, resulting in potential differences in their efficiency for the purpose of cell cycle synchronization. The exact same issues also extend to the DNA polymerase, Eg5 , and proteosome inhibitors.*

In the revised manuscript, we have included additional inhibitors to evaluate their effectiveness in cell synchronization. Additional inhibitors included flavopiridol (Cdk1 inhibitor), BMS-265246 (Cdk1 inhibitor), monastrol (Eg5 inhibitor), serum starvation, double thymidine block, ribociclib (Cdk4/6 inhibitor), abemaciclib (Cdk4/6 inhibitor), and hydroxyurea (HU). G1 Phase Synchronization: Treatment with 2 μ M ribociclib for 24 hours arrested 92% of cells in the G1 phase, though a small fraction remained in the S and G2 phases. In contrast, palbociclib demonstrated superior efficacy, achieving complete G1 phase arrest (100%) at concentrations >0.1 μ M. Treatment with 2 μ M abemaciclib resulted in 73% of cells arresting in G1, further emphasizing palbociclib's enhanced performance among Cdk4/6 inhibitors. S Phase Synchronization: HU (2 mM for 24 hours) increased the proportion of S-phase cells to 51%, with 45% specifically in early S-phase. However, its synchronization efficiency was notably lower compared to aphidicolin washout. Prometaphase synchronization: both 2 μ M and 4 μ M monastrol arrested approximately 60% of cells in prometaphase, a performance comparable to STLC but with slightly higher efficacy. While monastrol achieves cell arrest at marginally lower concentrations than STLC, it is significantly more expensive. G2 phase Synchronization: flavopiridol, at concentrations ranging from 2 to 10 μ M, was ineffective in arresting cells in the G2 phase. Treatment with 5 μ M BMS-265246 arrested approximately 41% of cells in G2, but its efficacy remained significantly lower than that of RO-3306. Notably, consistent with RO-3306

treatment, nuclear bubbling was observed in cells treated with BMS-265246.

Both the double thymidine block and serum starvation methods are discussed in detail in the subsequent section. All new results have been incorporated into the revised manuscript and are presented in Supplementary Figures S1C-D, S2D-E, S3A-C, S5C-D, S7A-D, and S8A-B.

2. The inhibitor-based synchronization strategies in comparison to traditional synchronization approaches like thymidine and serum starvation should be discussed.

Thank you for raising this important point. In the revised manuscript, we have incorporated traditional synchronization methods and quantified cell cycle stages using the same methodology. For serum starvation, cells were incubated in FBS-free media for 48 hours, as 24 hours was insufficient to induce the G0 phase. This treatment resulted in ~80% of cells entering the G0 phase. Upon incubation in serum-containing media, the proportion of G0 phase cells decreased to ~50% after 24 hours and ~20% after 48 hours. These results confirm that serum starvation effectively synchronizes cells in the G0 phase; however, these G0 cells exhibit slower re-entry into the cell division cycle following serum reintroduction. The new results are presented in Supplementary Figure S3.

For the thymidine block, we initially evaluated cell cycle stage distribution after 6 and 12 hours of single thymidine treatment (2 mM). As expected, no significant changes were observed after 6 hours, while S-phase cells increased to ~70% at 12 hours. Double thymidine treatment (see Methods) arrested ~72% of cells in the G1 phase. However, compared to palbociclib treatment, a notable proportion of cells (~28%) remained in the S and G2 phases. Following washout from the double thymidine block, most cells resumed cell cycle progression, but ~30% remained in the G1 phase, similar to palbociclib treatment. In our experiments, palbociclib was more effective and time-efficient than double thymidine treatment for synchronizing cells in the G1 phase. The new results are presented in Supplementary Figure S1C-D.

Reviewer #2 (Comments to the Authors (Required)):

This is overall a study that is well done and the table does a good job summarizing strategies to use for cell synchronization. There is a small number of suggestions to improve the paper. This would just need some further data mining from the existing data sets (live imaging).

The Palbociclib release experiment (Fig 1c) seems to be showing that resumption of cell cycle at all concentrations is not so good. I would expect most of the cells to reach mitosis and become arrested during this 18 hr release. Or, are cells slipping out of mitosis. The live imaging of this washout experiment could be quantified in terms of mitotic index. That would reveal the kinetics and if cells

are slipping out of mitosis.

Thank you for highlighting this important point, and we sincerely apologize for any confusion it may have caused. To clarify, we included STLC, an Eg5 inhibitor, in our majority of washout experiments, including those presented in Figure 1C and 1D (both fixed and live-cell imaging), to prevent mitotic exit. Our data confirm that after approximately 24 hours of STLC treatment, the vast majority of RPE1 cells remained arrested in mitosis (Figure 4A–D), indicating minimal mitotic slippage across all conditions. In the revised manuscript, we have addressed this point by explicitly including this information in the corresponding figure legends and Method section.

The live imaging in the aphidicolin experiment (Fig 2d) could be quantified in terms of cumulative mitotic index to reveal the kinetics of cells arriving at mitosis. That would be valuable.

Figure 3D highlights cells specifically in the S phase, whereas subsequent panels, Figure 2E (5 $\mu\text{g/ml}$) and Supplementary Figure S5B (2.5 $\mu\text{g/ml}$ and 10 $\mu\text{g/ml}$), encompass all cell cycle stages, including G2 and mitosis. At 4 and 6 hours post-aphidicolin washout, across all tested concentrations (2.5–10 $\mu\text{g/ml}$), no mitotic cells were observed. However, a small population of G2 phase cells (less than 5% of the total, comprising both early and late G2 phases) was detected. These results align with the live-cell aphidicolin washout experiment, which began to show mitotic cells approximately 9 hours after the aphidicolin washout (Fig. 2F).

In the RO washout experiment (Fig 3d) only about half the G2 cells entered mitosis within 2 hrs. This seems quite poor recovery.

In the revised experiments, we implemented more extensive washout conditions to minimize the underestimation of mitotic cells following RO-3306 washout. Cells were washed five times with complete growth medium supplemented with 10% FBS. Additionally, we introduced a 2x concentration of PFA directly into the growth medium containing STLC, followed by immediate DNA staining without performing immunostaining or additional wash steps to minimize removal of mitotic cells from coverslips. Under these conditions, cells treated with 3 μM and 6 μM RO-3306 exhibited approximately 35% mitotic cells at 2 hours post-washout. In contrast, cells treated with 1 μM RO-3306 showed a mitotic index of 14%, while those treated with 10 μM RO-3306 displayed no detectable mitotic cells (Figure 3E). Notably, cells treated with 3 μM RO-3306 demonstrated an increased mitotic index under these conditions, whereas the results for 6 μM RO-3306 were consistent with those reported in the original manuscript (~35% mitotic cells). These findings suggest that treatments with both 3 μM and 6 μM RO-3306 facilitate the progression of approximately 60% of cells into mitosis within 2 hours post-washout. The updated results have been incorporated into the revised manuscript.

In Fig 3e, 2 hrs after washing out the 6 μ M RO into STLC medium, only 8% of cells were in mitosis. The shows lack of recovery. However, in Fig 3i, there seem to be about 70% cells in mitosis in the 15-29 minute time points. What is the explanation for this difference?

In the original experiments, it appears that we lost some mitotic cells in the fixed samples for Figure 3E, particularly under the 3 μ M RO-3306 treatment condition (see above section). We believe the lower estimation can be attributed to two main factors. First, mitotic cells are generally more prone to detachment compared to interphase cells due to their rounded morphology. Second, 3 μ M RO-3306 likely represents a borderline concentration that effectively arrests most cells in the G2 phase, but a subset proceeds into mitosis without washout. These cells display unique bubbled nuclei, which were not observed in cells treated with 6 μ M RO-3306. Consequently, cells treated with 3 μ M RO-3306 appeared more stressed and detached more easily, even in the non-mitotic population, potentially contributing to the lower estimation of the mitotic index under this condition. To address this issue in the revised manuscript, we fixed cells with PFA directly supplemented into the growth medium (details provided above). Using this approach, we observed a comparable mitotic index of approximately 35% in both 3 μ M and 6 μ M RO-3306-treated cells (Fig. 3E).

Additionally, we reassessed live-cell imaging for RO-3306 washout experiments. Most of our studies employed RPE1 H2B-EGFP stably expressing cell lines for live-cell imaging, which allowed us to minimize phototoxicity that could affect cell cycle progression. However, this cell line, while derived from the same RPE1 parental cell type used in fixed-cell assays, represents a different clonal population. In the revised manuscript, we re-performed live-cell imaging using the exact same RPE1 cells used for the fixed-cell experiments. For these experiments, we labeled DNA with Nuc650 at least one hour prior to imaging. Two distinct conditions were tested: (1) using complete growth media for the washout, allowing cells to progress into anaphase, and (2) using complete growth media supplemented with 5 μ M STLC, which arrests cells in prometaphase after entering mitosis. The results were largely consistent with those obtained using the RPE1 H2B-EGFP-expressing cells. Most cells exhibited NEBD within ~23 minutes after RO-3306 washout, with a slightly faster onset in cells treated with 3 μ M RO-3306 (new Fig. 3G). These cells progressed to anaphase onset around 60 minutes post-washout (new Fig. 3H). Cells treated with 6 μ M RO-3306 showed a slight delay in NEBD compared to those treated with 3 μ M RO-3306 (new Fig. 3I). We also quantified the mitotic index at 2 hours post-RO-3306 washout in live-cell imaging under STLC-supplemented conditions. The mitotic index ranged from 20–38% in treated cells, while control asynchronous cells incubated with STLC for 2 hours showed only a 4% increase in mitotic cells (Fig. 3J). Notably, the mitotic index at 2 hours post-washout with 6 μ M RO-3306 was consistent with the fixed-cell results. However, the 3 μ M RO-3306 washout in live-cell imaging exhibited a slightly

lower mitotic index (22%) compared to the ~35% observed in fixed-cell assays. All washout live-cell experiments were conducted simultaneously using multi-well chamber slides. It is worth noting that under 3 μ M RO-3306 washout conditions, cells were more prone to detachment compared to other conditions, including control, 6 μ M, and 10 μ M RO-3306-treated cells. This likely led to a slight underestimation of the mitotic index in this specific condition.

Minor comments.

1. The title is a bit vague. How about something like, "Reversible and effective synchronization methods for studying stage-specific cell cycle processes."

Thank you for your suggestion. We have updated the title accordingly.

Reviewer #3 (Comments to the Authors (Required)):

In this manuscript, the authors used several small molecule inhibitors to synchronize human nontransformed RPE1 cells in various phases of the cell cycle. Not surprisingly, they observed that treatment with CDK4/6 inhibitor synchronized cells in G1 phase, aphidicolin at G1/S transition, CDK1 inhibitor in G2 and Eg5 inhibitor in prometaphase. These phenotypes correspond well to the known modes of action of all these molecules. Also not surprisingly, they observed that lower concentrations were usually not as efficient as optimal concentrations of the drugs. On the other hand, too high concentrations showed suboptimal synchronization and/or extended recovery times probably due to the lower specificity towards the expected targets. I am somewhat confused how to evaluate this study as all the observed effects have been reported previously although maybe not in such systematic way. These protocols may be handy for researchers outside from the cell cycle field but in my opinion, they do not bring any new view of the synchronization of the cells. The doses of the compounds were optimized specifically for RPE1 cells, which are widely used in the field. Nevertheless the values are not universally valid for other cell types so similar optimization procedures will be necessary for other cellular models. In the summary, I appreciate the effort of authors with careful optimization of their experimental system but I feel that online protocol might be more suitable form of publication.

This study highlights the impact of commonly used and more recently developed cell synchronization methods targeting various stages of the cell cycle. By leveraging an immunofluorescence microscopy-based assay, it provides high-resolution insights into their effects on the cell cycle, which are challenging to achieve with population-based accuracy in flow cytometry assays. While the study is limited to RPE1 cell lines, its findings offer valuable insights for broader research on cell cycle regulation. Furthermore, the optimized method

established for RPE1 cells can serve as a benchmark for standardizing protocols in other cell types.

Reviewer #4 (Comments to the Authors (Required)):

Asynchronously-proliferating cell populations have cells in different cell cycle phases. Analyzing factors that oscillate in the cell cycle can thus be challenging to interpret. Thus, effective synchronization methods are helpful to address this challenge.

In this manuscript, Chen et al., provide a comprehensive guide to synchronization by cell cycle block and release methods for human RPE1 cells using drug inhibitors. They highlight that specific drug concentrations could be effective in synchronizing cells but could also affect normal cell cycle progression post-release. The authors admirably acknowledge that the recommendations they give in this manuscript apply for untransformed RPE1 cells, and may not be applicable for other cell lines especially cancer cell lines, though it is a starting point for extrapolating to other cell lines. This paper will be of interest to readers who intend to use pharmacological inhibitors for RPE1 synchronization and release.

The methods for analyzing cell cycle phase in this study lean heavily on a second study by the same group which is available as a pre-print "ImmunoID...".

We support publication if authors address these major points:

Thank you for recognizing the significance of our work. We believe the results presented in this paper will not only be valuable but also serve as a standard for optimization in other cell types. We sincerely appreciate the reviewer for highlighting the critical issue regarding aphidicolin. Without this insightful feedback, we might have drawn incorrect conclusions. The details of our responses have been thoroughly addressed in the point-by-point section. Furthermore, the application of ImmunoCellCycle-ID in this study enabled us to assess the precise effects of cell cycle synchronization methods on distinct phases of the cell cycle. We apologize for the earlier difficulty in accessing the full dataset of the ImmunoCellCycle-ID paper. In the revised manuscript, a PDF copy of the paper, including supplementary figures, has been provided for clarity and completeness.

1) The PCNA protein is expressed throughout the cell cycle, and it is not in fact cell cycle-regulated at the level of total protein. The figures as presented give the impression that total PCNA is cell cycle-regulated. On the other hand, PCNA localizes to DNA detectable as foci. Small foci appear in early S phase, and more discrete foci appear in late S phase. It is shown in multiple images that the PCNA signal is not present in G1 cells although it should be if cells were not pre-extracted. Authors mention in the methods section using PFA or methanol for fixation. However, it is not clear which

fixation method is used for which protein targets, and whether the signal detected is for soluble proteins or chromatin bound proteins. Based on the data shown here (and in the accompanying pre-print) we assume the PCNA staining was with methanol fixation which would remove soluble protein. Unfortunately, the relevant information is not available in the ImmunoID paper because the supplemental files aren't posted by bioRxiv.

We sincerely apologize that the reviewer was unable to access the supplementary files for the Immuno-CellCycle-ID paper. The paper is now published in Journal of Cell Science (DOI: 10.1242/jcs.263414) and includes a detailed method for identifying sub-stages of the cell cycle using both qualitative and minimal quantitative approaches. Specifically: For four-color staining (CENP-F, CENP-C, PCNA, and DNA), sub-stages can be identified without quantification. For three-color staining (excluding either CENP-C, CENP-F, or PCNA), minimal quantification is sufficient. The manuscript also details the changes in PCNA signals during the G1-to-S phase transition. Briefly: PCNA marks the nucleus from G1 to G2 phases but exhibits distinct foci during S phase, which evolve as S phase progresses. These changes allow differentiation between early and late S phases. However, PCNA staining alone cannot distinguish G1 from G2 phases, as both exhibit nearly identical nuclear signals. To address this, the following staining combinations are effective: CENP-F and PCNA: These together can distinguish G1 from G2 phases without requiring quantification. CENP-F signals are only present from S to G2 phases. CENP-C without PCNA: CENP-C can differentiate G1 from G2 phases based on kinetochore signals. In early G2, kinetochore signals are twice as bright as in G1, and by late G2, kinetochores form pairs. It is worth noting that all tested PCNA antibodies work exclusively with MeOH fixation, whereas CENP-F and CENP-C antibodies are compatible with both PFA and MeOH fixation. As MeOH fixation has a higher risk of losing mitotic cells compared to PFA fixation, we have adopted a flexible approach, selecting fixation methods appropriate for each experiment. The manuscript has now been revised to include detailed methods for each experimental condition and a PDF copy of ImmunoCellCycle-ID paper has been included with the revision.

2) Moreover, it is not clearly explained in the text here (or the ImmunoID paper) which proteins are cell cycle regulated or their general mechanism of regulation with citations. For example, CENP-F behaves like an APC/C substrate; is that so? Authors should clarify these points in each figure legend or in text.

Thank you for raising this important point. In our ImmunoCellCycle-ID paper, we demonstrated that a combination of labeling CENP-F, PCNA, CENP-C, and DNA can distinguish detailed sub-stages of the cell cycle without requiring extensive quantification. Even in the absence of CENP-F or CENP-C, accurate staging can be achieved with minimal

quantification, as detailed in the paper. Although CENP-F is commonly known as a kinetochore protein, our findings, along with those of other groups, reveal that it predominantly localizes to the nucleus, not the kinetochores, during the S and G2 phases. It transitions to the kinetochores starting in prophase. Additionally, previous studies have suggested that CENP-F serves as a substrate for both Cdc20 and the APC/C, leading to its degradation post-anaphase (Gurden et al., 2010, *Journal of Cell Science*, DOI: 10.1242/jcs.062075). However, the mechanism underlying its nuclear recruitment during the S phase remains unclear, and further investigations are required to elucidate the precise details of its nuclear localization. We have revised the Methods section (Cell cycle stage identification) to incorporate these details.

3) For synchronizing cells with aphidicolin (Fig. 2), authors argue that aphidicolin effectively inhibits DNA synthesis and arrests RPE1 cells in G1 phase. We disagree that the data support this conclusion, and aphidicolin does not prevent origin firing, just fork progression. In Fig. 2a, small PCNA foci start to appear in the arrested cells. Similarly, CENP-F is shown to be expressed in early S in Fig. 2a, while in suppl Fig. 1, it is depicted as a late S phase marker. We suggest to authors to stain for other G1 or S phase biochemical markers - besides PCNA - to strengthen their claim. For example: CDT1 should be still present if cells are truly arrested in G1 phase without origin firing. If origins have begun firing but forks arrested by aphidicolin, then S phase markers will appear: increase in cyclin A and geminin, loss of p21 and CDT1, etc. PCNA itself is not a sensitive enough measure at this resolution to distinguish very early S from late G1.

We sincerely appreciate the reviewer for raising this critical point. PCNA is one of the most commonly used markers to determine the S phase, as it exhibits distinctly different nuclear localization patterns compared to those in the G1 and G2 phases, as detailed in our recent study (Chen YL et al., *Journal of Cell Science*, 2024; a PDF copy of this paper has been included with the revision). In response to the reviewer's comment, we carefully re-assessed the impact of aphidicolin on cell cycle stages. In the original manuscript, we concluded that aphidicolin arrests cells in the G1 phase based on the absence of PCNA puncta, which are characteristic of the S phase. However, a significant proportion of these cells exhibited nuclear CENP-F signals. While such signals (no PCNA nuclear puncta with CENP-F signals) are typically associated with the G2 phase, the CENP-F staining in aphidicolin-treated cells appeared weaker and more comparable to that seen in the S phase. To further clarify the true cell cycle stage of these cells, we employed two additional markers: phosphorylated Rb (pRb) and BRCA1. BRCA1 forms distinct nuclear puncta from early S phase, intensifies during late S phase, and gradually diminishes in early G2 phase. On the other hand, pRb nuclear signals markedly increase from early S phase to prometaphase (Chen YL et al., *Journal of Cell Science*, 2024). Interestingly, while PCNA signals suggested that aphidicolin-treated cells were arrested in the G1 phase, analyses using pRb, BRCA1, and CENP-F indicate that these cells are likely in the S phase. In

the revised manuscript, we have provided detailed results of pRb and BRCA1 staining and updated our conclusions to reflect that aphidicolin effectively arrests cells in the S phase. We have also noted the unique scenario in which PCNA is not an appropriate marker for this specific case.

4) The authors note that ~30% don't recover from the lower dose of Palbociclib. Most cell lines spontaneously enter quiescence. What fraction of proliferating cells in this study are quiescent?

To address this important point, we developed a method for accurately identifying G0-phase cells through the dual labeling of PCNA and acetylated tubulin. The methodology is detailed in ImmunoCellCycle-ID (Chen YL et al., Journal of Cell Science, 2024, Supplementary Figure 5). Using this approach, we quantified the G0-phase population following 24-hour treatment with Palbociclib (0.25 and 1.0 μ M) and 24 hours after Palbociclib washout. In asynchronous RPE1 cells, only 2% were in the G0 phase. G0 phase cells increased to 9–13% after 24 hours of palbociclib treatment and subsequently decreased to 4–7% following Palbociclib washout. These findings indicate that approximately 10% of G1-arrested cells induced by Palbociclib enter the G0 phase, and a subset of these cells remains in the G0 phase even after the washout. The new data are presented in Supplementary Figure S2B-C.

5) The authors should discuss the potential effects of arrest on subsequent cell behavior that are direct consequences of long-term arrest. Release from an early S phase block will be early S by DNA replication but late S by some molecular markers such as APC/C substrates which accumulated to high levels during the arrest. Stress responses may also contribute to cell cycle progression for some time after release.

Thank you for highlighting this crucial point. A primary objective of this study is to achieve efficient synchronization of the majority of cells at the target cell cycle stage with minimal intervention. Based on our optimization, we chose a 24-hour incubation period for most inhibitors. This duration was selected because incubation times shorter than 12 hours were insufficient for effective synchronization with most inhibitors, while extending treatment to approximately 48 hours did not significantly enhance synchronization efficacy. We fully acknowledge that prolonged incubation, even for 24 hours, may introduce artificial stress responses or subtle defects not typically observed during normal cell cycles. To address this concern, we have included the following statement in the 'Limitations of this Study' section: ' To efficiently synchronize most cells at specific cell cycle stages with minimal process, we incubated cells with most inhibitors for 24 hours in this study. However, this prolonged incubation may induce artificial stress responses or subtle, undetectable defects in our assay, following the washout. Furthermore, our investigation into drug recovery focused on the short-term effects

following drug release, with the possibility of uncovering additional defects in subsequent cell cycle phases.”

Minor points for the authors' consideration:

- *Line 59: "reproductive cell cycle" can mean something else other than the cell division cycle.*

No, this refers exclusively to the cell division cycle, excluding the resting phase (G0). We have revised the sentence accordingly.

- *Line 102: "presented in this study"*

We have corrected.

- *Line 171: The right-hand side panel of Fig. 2b is not clear. What is the x-axis depicting?*

It represents the frequency of cells in the S phase (%) at 4 and 6 hours post-aphidicolin washout (denoted as 4 hrs post-wo and 6 hrs post-wo, respectively). To enhance clarity, we have updated the labels on both the x- and y-axes.

- *Line 171: Suppl. Fig 2c: at the 10 ug/mL concentration without washout, cells are depicted as G1 cells although CENP-F (S phase marker) and PCNA is expressed (same concern about bound versus unbound proteins).*

Thank you very much for identifying this critical error. We have now corrected the image.

- *Line 231: Please clarify that 3 μ M and higher concentrations $<10 \mu$ M of RO-3306 can efficiently arrest most cells in G2 because the presented data in this paper show that 10 μ M arrest cells in G1 and might inhibit other CDKs, as explained by the authors.*

In our results with RPE1 cells, 10 μ M of RO-3306 was ineffective to arrest cells in G2, instead resulting in arrest in G1. We have revised the sentence for precision as follows: 'These results demonstrate that RO-3306 at concentrations of 3 and 6 μ M effectively arrests the majority of cells in the G2 phase, whereas concentrations of 1 μ M and 10 μ M fail to induce this effect. Notably, a subset of G2-arrested cells progresses into mitosis following the washout of RO-3306.'

- *Line 254: please spell out Nuclear Envelope Breakdown (NBD)*

We have corrected.

- *Line 266: "as expected" instead of "as expect".*

We have corrected.

- *Line 352: Limitations section: It could be helpful to readers to point out that they focused in this paper on short-term recovery post drug washout, and that longer release periods might reveal additional defects in subsequent cell cycles.*

Thank you for important suggestions. We have revised and added a sentence to read “Additionally, our investigation into drug recovery focused on the short-term effects following drug release, with the possibility of uncovering additional defects in subsequent cell cycle phases.” in Limitations section.

February 10, 2025

RE: Life Science Alliance Manuscript #LSA-2024-03000R

Dr. Aussie Suzuki
University of Wisconsin-Madison
1111, Highland Ave, WIMR, Room 6533, UW-Madison
Madison, WI 53705

Dear Dr. Suzuki,

Thank you for submitting your revised manuscript entitled "Reversible and effective cell cycle synchronization method for studying stage-specific processes". We would be happy to publish your paper in Life Science Alliance pending final revisions necessary to meet our formatting guidelines.

- please address the Reviewers' remaining comments
- please be sure that the authorship listing and order is correct
- please add the X and Bluesky handle of your host institute/organization as well as your own or/and one of the authors in our system
- please add your main, supplementary figure, and table legends to the main manuscript text after the references section

LSA now encourages authors to provide a 30-60 second video where the study is briefly explained. We will use these videos on social media to promote the published paper and the presenting author (for examples, see <https://docs.google.com/document/d/1-UWCfbE4pGcDdcgzcmiuJl2XMBJnxKYeqRvLLrLSo8s/edit?usp=sharing>). Corresponding or first-authors are welcome to submit the video. Please submit only one video per manuscript. The video can be emailed to contact@life-science-alliance.org

A. FINAL FILES:

B. MANUSCRIPT ORGANIZATION AND FORMATTING:

Thank you for your attention to these final processing requirements. Please revise and format the manuscript and upload materials within 14 days.

Sincerely,

Reviewer #1 (Comments to the Authors (Required)):

The author effectively addressed the concerns, and the manuscript became an invaluable resource for researchers who employed the RPE1 cell line model. One significant point to add to the manuscript is that the activation of p53 in various synchronization methods has not been studied and may be a crucial component in efficient synchronization.

Reviewer #2 (Comments to the Authors (Required)):

My concerns have been addressed.

Reviewer #4 (Comments to the Authors (Required)):

Thank you for your revisions. We regard the following points as crucial for the clarity of this manuscript and for our satisfaction with this work:

1) The authors explained to us in their response the dynamics of PCNA during the cell cycle, which we certainly are aware of. However, they still have not explained to readers whether the detected signals for several protein targets (including PCNA) correspond to insoluble/ bound proteins. The new methods section does not in fact say which fixation was used for each antigen. Methanol fixation also permeabilizes cells which removes soluble PCNA leaving only the DNA-bound S phase PCNA foci, so it is actually quite important to know when methanol fixation was used. Now that we know PCNA staining was on methanol-fixed cells, we are quite certain their PCNA signal is only the bound protein and not total. Please specifically clarify this in the text of the manuscript. We consider this clarification quite important to avoid confusion in the literature about exactly what is being measured. Furthermore, given that the authors understand and agree that aphidicolin causes early S and not G1 arrest, they should relabel some of their "G1" cells that have low PCNA signal after methanol extraction in aphidicolin as "early S" instead.

2) The authors explained to us the cell cycle regulation of CENP-F, provided citations, and they added the paragraph "Cell cycle stage identification" to the methods section. However, our original request was not to explain to "us" which target was used as a marker of which cell cycle stage because that was clear. We asked the revision to explain to readers the cell cycle regulation of the studied protein targets in the manuscript (with citations). This regulation is the basis for their cell cycle phase identification; just citing their other (subscription-protected) paper is not fair to readers.

3) The authors assert that phosphorylated RB (pRB) signal is not detected before the start of S phase, but this is not true in actively proliferating cells. The well-documented late G1 RB phosphorylation is now known to be only applicable to cells that are returning to G1 from quiescence. Textbooks still extrapolate that these dynamics apply during all G1 phases. However, the Spencer and Meyer labs have extensively studied pRB regulation during the proliferative cell cycle (PMID: 28514656, PMID: 24075009 among others), and have shown that Rb is phosphorylated right after mitosis in G1 phases of proliferating cells, and it is only absent in G0 (quiescent) RPE1 cells. (We are not affiliated with these labs, but we have made similar observations in

RPE cells that are unpublished.) Spencer lab papers have established that a subset of cells enter spontaneous quiescence even under optimum proliferation conditions in culture. The probability of spontaneous quiescence varies by cell line. This contradicts the authors' statement about pRB detection only at the start of S phase, and the authors should fix their interpretation.

Minor: consider replacing "reproductive cycle" with "cell division cycle" to avoid potential confusion with meiosis (line 488). The authors have changed the term in one sentence but not on line 488.

We notice a different field of cells is shown for the new Supplementary figure 5 than was in the original figure Supplementary figure 2.

Point-by-point response

Reviewer #1 (Comments to the Authors (Required)):

The author effectively addressed the concerns, and the manuscript became an invaluable resource for researchers who employed the RPE1 cell line model. One significant point to add to the manuscript is that the activation of p53 in various synchronization methods has not been studied and may be a crucial component in efficient synchronization.

We fully agree with this point. Although we addressed this in the Limitations of the Study section, we have further emphasized it in the revised manuscript as follows (changes are highlighted in yellow): “For our synchronization method, we optimized the protocol using the RPE1 cell line, a normal, non-transformed human cell line expressing wild-type p53 (Bowden et al, 2020). It has been reported that certain inhibitors, particularly Cdk inhibitors, exhibit varying efficacies across different cell lines (Johnson et al., 2021; Trotter & Hagan, 2020). This variability may be attributed to the differential activities of Cdks in distinct cell types. A study demonstrated that in cancer cells, Cdk2 can compensate for the loss of Cdk1 during mitotic entry when Cdk1 is rapidly degraded using the auxin-degron system (Lau et al., 2021). However, this compensation does not occur in normal cells. While our optimized inhibitor concentrations serve as a valuable reference for cell cycle synchronization, adjustments may be necessary when applied to other cell lines, especially those with p53 deletions or mutations....”

Reviewer #4 (Comments to the Authors (Required)):

Thank you for your revisions. We regard the following points as crucial for the clarity of this manuscript and for our satisfaction with this work:

1) The authors explained to us in their response the dynamics of PCNA during the cell cycle, which we certainly are aware of. However, they still have not explained to readers whether the detected signals for several protein targets (including PCNA) correspond to insoluble/ bound proteins. The new methods section does not in fact say which fixation was used for each antigen. Methanol fixation also permeabilizes cells which removes soluble PCNA leaving only the DNA-bound S phase PCNA foci, so it is actually quite important to know when methanol fixation was used. Now that we know PCNA staining was on methanol-fixed cells, we are quite certain their PCNA signal is only the bound protein and not total. Please specifically clarify this in the text of the manuscript. We consider this clarification quite important to avoid confusion in the literature about exactly what is being measured. Furthermore, given that the authors understand and agree that aphidicolin causes early S and not G1 arrest, they should relabel some of their "G1" cells that have low PCNA signal after methanol extraction in aphidicolin as "early S" instead.

In the revised manuscript, we have included information on which experiments utilized MeOH or PFA fixation. CENP-F and CENP-C antibodies perform comparably well with both fixation methods. However, detecting nuclear punctate PCNA requires MeOH fixation, likely due to the reason you suggested, that the removal of soluble PCNA is necessary to visualize DNA-bound S-phase PCNA foci. We have tested multiple PCNA antibodies but all of them required MeOH fixation (Chen et al., JCS, 2024). Consequently, all immunofluorescence imaging involving PCNA was performed using MeOH fixation in this study. However, MeOH fixation carries the risk of underestimating the mitotic population, as it has a higher likelihood of losing rounded mitotic cells. Therefore, for experiments focused on mitotic cell populations, we primarily used PFA fixation, although we validated most of these experiments with MeOH fixation as well.

We sincerely appreciate your suggestion in the first revision regarding the potential limitations of PCNA as a marker for aphidicolin treatment. This insight allowed us to explore additional details and incorporate alternative markers, such as BRCA1 and pRb, into our analysis for aphidicolin condition. We observed that low, uniform nuclear PCNA signals (rather than specific foci) are present in all interphase cells during G1 and G2 phases, making PCNA unsuitable for distinguishing between these two phases (Chen et al., JCS, 2024). In aphidicolin-treated cells, all cells exhibited uniform PCNA staining, similar to G1 or G2 phase cells in asynchronous RPE1 populations. Therefore, we determined the G1 or S phase status based on BRCA1+CENP-F, pRb+CENP-F, or CENP-C levels combined with CENP-F. The cells identified as G1 phase in Figure 2A and Supplementary Figure 5B lacked CENP-F and showed no BRCA1 or pRb signal. It is important to note that the regulation of CENP-F nuclear transport remains unknown, though it appears to be tightly coupled to S-phase entry and independent of PCNA-mediated control.

2) The authors explained to us the cell cycle regulation of CENP-F, provided citations, and they added the paragraph "Cell cycle stage identification" to the methods section. However, our original request was not to explain to "us" which target was used as a marker of which cell cycle stage because that was clear. We asked the revision to explain to readers the cell cycle regulation of the studied protein targets in the manuscript (with citations). This regulation is the basis for their cell cycle phase identification; just citing their other (subscription-protected) paper is not fair to readers.

We have revised the manuscript to provide a more detailed description of the ImmunoCellCycle-ID methods, incorporating relevant references on CENP-F, PCNA, and CENP-C.

3) The authors assert that phosphorylated RB (pRB) signal is not detected before the start of S phase,

but this is not true in actively proliferating cells. The well-documented late G1 RB phosphorylation is now known to be only applicable to cells that are returning to G1 from quiescence. Textbooks still extrapolate that these dynamics apply during all G1 phases. However, the Spencer and Meyer labs have extensively studied pRB regulation during the proliferative cell cycle (PMID: 28514656, PMID: 24075009 among others), and have shown that Rb is phosphorylated right after mitosis in G1 phases of proliferating cells, and it is only absent in G0 (quiescent) RPE1 cells. (We are not affiliated with these labs, but we have made similar observations in RPE cells that are unpublished.) Spencer lab papers have established that a subset of cells enter spontaneous quiescence even under optimum proliferation conditions in culture. The probability of spontaneous quiescence varies by cell line. This contradicts the authors' statement about pRB detection only at the start of S phase, and the authors should fix their interpretation.

Thank you for your valuable suggestions. We agree with this point. In asynchronous RPE1 cells, we found that approximately 90% of pRb-positive cells (characterized by nuclear pRb signals) exhibited either CENP-F nuclear signals (without punctate PCNA) or both nuclear punctate PCNA and CENP-F nuclear signals. Based on the references provided, we concluded that the remaining ~10% of pRb-positive cells lacking both CENP-F and punctate PCNA nuclear signals correspond to the late G1 phase. Interestingly, in aphidicolin-treated cells, we did not observe any pRb-positive populations lacking both PCNA and CENP-F signals, further supporting that aphidicolin arrests most cells in S phase. In this study, we did not determine S phase solely based on pRb or BRCA1 nuclear signals; instead, we used pRb or BRCA1 in combination with CENP-F, as PCNA becomes less reliable for S-phase identification under aphidicolin treatment.

We have incorporated the suggested references and revised the relevant section of Results as follows (changes are highlighted in yellow): “To further validate these results, we assessed pRb, PCNA, and CENP-F staining. Our previous study demonstrated a significant increase in nuclear pRb signals from early S phase (Figs. S4C-E) (Chen et al., 2024b). **In asynchronous RPE1 cells, we found that ~90% of pRb-positive cells exhibited either exclusive nuclear CENP-F signals (indicating G2 phase) or both punctate nuclear PCNA and CENP-F signals (indicating S phase). pRb-positive cells lacking both PCNA nuclear puncta and nuclear CENP-F signals were likely in late G1 phase (Arora et al, 2017; Spencer et al, 2013), however, they comprised only ~10% of the total interphase population.** Similar to BRCA1, ~80% of pRb-positive cells in asynchronous RPE1 cells displayed punctate nuclear PCNA and nuclear CENP-F signals, indicating them as S phase cells, while ~20% exhibited CENP-F nuclear signals without PCNA puncta, suggesting G2 phase. Notably, in cells treated with aphidicolin, the majority of pRb-positive cells (~78%) lacked punctate PCNA nuclear signals but exhibited weaker CENP-F nuclear signals compared to typical G2 phase cells, indicating that these cells were in S phase (Figs. S4C-E). **Unlike asynchronous cells, no pRB-positive cells lacking both**

PCNA and CENP-F signals were observed. In summary, based on pRb and CENP-F staining, approximately 87% of cells were arrested in S phase after 24 hours of treatment with 5 µg/ml aphidicolin. Furthermore, based on CENP-F and BRCA1 or pRb staining, we concluded that aphidicolin at concentrations of 2.5, 5, and 10 µg/ml effectively arrested cells in the S phase likely near the G1/S boundary, achieving an arrest rate of 70–80% (Figs. 2A-B).”

Minor:

4) consider replacing "reproductive cycle" with "cell division cycle" to avoid potential confusion with meiosis (line 488). The authors have changed the term in one sentence but not on line 488.

Thank you for your suggestion. We have revised the wording accordingly.

5) We notice a different field of cells is shown for the new Supplementary figure 5 than was in the original figure Supplementary figure 2.

Apologies for the oversight. Yes, we have updated only the control images for aphidicolin-treated cells (2.5 and 10 µg/ml) at the 0-hour time point in Supplementary Figure 5B (top panel) to provide clearer representative locations corresponding to the quantification of S-phase arrest.

February 21, 2025

RE: Life Science Alliance Manuscript #LSA-2024-03000RR

Dr. Aussie Suzuki
University of Wisconsin-Madison
1111, Highland Ave, WIMR, Room 6533, UW-Madison
Madison, WI 53705

Dear Dr. Suzuki,

Thank you for submitting your Methods entitled "Reversible and effective cell cycle synchronization method for studying stage-specific processes". It is a pleasure to let you know that your manuscript is now accepted for publication in Life Science Alliance. Congratulations on this interesting work.

DISTRIBUTION OF MATERIALS:

Again, congratulations on a very nice paper. I hope you found the review process to be constructive and are pleased with how the manuscript was handled editorially. We look forward to future exciting submissions from your lab.

Sincerely,
